# Optimal Graph Clustering
# without Edge Density Signals

**Maximilien Dreveton**
EPFL
maximilien.dreveton@gmail.com

**Elaine S. Liu**
EPFL
elaineliu@stanford.edu

**Matthias Grossglauser**
EPFL
matthias.grossglauser@epfl.ch

**Patrick Thiran**
EPFL
patrick.thiran@epfl.ch

## Abstract

This paper establishes the theoretical limits of graph clustering under the Popularity-Adjusted Block Model (PABM), addressing limitations of existing models. In contrast to the Stochastic Block Model (SBM), which assumes uniform vertex degrees, and to the Degree-Corrected Block Model (DCBM), which applies uniform degree corrections across clusters, PABM introduces separate popularity parameters for intra- and inter-cluster connections. Our main contribution is the characterization of the optimal error rate for clustering under PABM, which provides novel insights on clustering hardness: we demonstrate that unlike SBM and DCBM, cluster recovery remains possible in PABM even when traditional edge-density signals vanish, provided intra- and inter-cluster popularity coefficients differ. This highlights a dimension of degree heterogeneity captured by PABM but overlooked by DCBM: *local* differences in connectivity patterns can enhance cluster separability independently of *global* edge densities. Finally, because PABM exhibits a richer structure, its expected adjacency matrix has rank between $k$ and $k^2$, where $k$ is the number of clusters. As a result, spectral embeddings based on the top $k$ eigenvectors may fail to capture important structural information. Our numerical experiments on both synthetic and real datasets confirm that spectral clustering algorithms incorporating $k^2$ eigenvectors outperform traditional spectral approaches.

## 1 Introduction

Graph clustering is the task of partitioning the vertex set of a graph into non-overlapping groups such that vertices within the same group exhibit similar patterns or properties. As a fundamental task in the statistical analysis of networks, graph clustering plays a key role in revealing the underlying structure and functional organization of complex networks (Avrachenkov and Dreveton, 2022).

Most graph clustering algorithms are based on the assumption that vertices within the same cluster are more densely connected than vertices in different communities. In other words, intra-cluster edge-density is higher than inter-cluster edge-density. Under this premise, metrics such as modularity, graph cuts, or their variants are commonly used to motivate and design graph clustering algorithms. However, these methods fundamentally rely on the edge density as their primary input signal. This leads to a natural question: *Is edge density essential for recovering clusters, or can other structural signals be exploited instead?* In this work, we demonstrate that the connection patterns of individual vertices can be exploited to recover clusters, even when intra-cluster and inter-cluster edge densities are equal.

39th Conference on Neural Information Processing Systems (NeurIPS 2025).

**Random graphs with cluster structure: block models with and without degree heterogeneity.**
Random graphs with cluster structure are often modeled using block models. Let $z \in [k]^n$ be a vector representing the cluster assignments of each vertex. For all the random graphs that we consider, the adjacency matrix $A \in \{0,1\}^{n \times n}$ is assumed to be symmetric with zero diagonal and $A_{ij} = A_{ji} \sim \mathrm{Ber}(P_{ij})$ for all $i > j$, where $P_{ij} \in [0,1]$ is the probability of an edge between vertices $i$ and $j$. The simplest block model supposes that

$$P_{ij} = \begin{cases} p & \text{if } z_i = z_j, \\ q & \text{otherwise.} \end{cases} \tag{1.1}$$

This model is often called the planted partition model, or the *stochastic block model* (SBM) with homogeneous interactions.[1] A known drawback of this model is that all vertices share the same expected degree. To mitigate this issue, Karrer and Newman (2011) proposed the *degree-corrected block model* (DCBM), where

$$P_{ij} = \begin{cases} \theta_i \theta_j p & \text{if } z_i = z_j, \\ \theta_i \theta_j q & \text{otherwise.} \end{cases} \tag{1.2}$$

The quantities $\theta_1, \cdots, \theta_n$ are the degree-correction parameters. To ensure identifiability, these parameters are normalized such that $\sum_{i:\, z_i = a} \theta_i = n_a(z)$ for all $a \in [k]$, where $n_a(z) = |\{i:\, z_i = a\}|$ denotes the size of cluster $a$.

However, the degree-correction parameter $\theta_i$ uniformly inflates or deflates the connection probabilities of vertex $i$ across all clusters. As a result, vertices with a large degree-correction parameter have more edges both within their own cluster and with other clusters. This makes it impossible to model vertices that exhibit higher connectivity exclusively within their own cluster. To mitigate this issue, Sengupta and Chen (2018) introduced the *popularity adjusted block model* (PABM), where

$$P_{ij} = \begin{cases} \lambda_i^{\text{in}} \lambda_j^{\text{in}} p & \text{if } z_i = z_j, \\ \lambda_i^{\text{out}} \lambda_j^{\text{out}} q & \text{otherwise.} \end{cases}$$

In this model, the quantity $\lambda_i^{\text{in}}$ (resp., $\lambda_i^{\text{out}}$) is the popularity of vertex $i$ with other vertices within its own cluster (resp., with vertices in other clusters). These coefficients are normalized such that $\sum_{i:\, z_i = a} \lambda_i^{\text{in}} = n_a(z)$ and $\sum_{i:\, z_i = a} \lambda_i^{\text{out}} = n_a(z)$ for all $a \in [k]$. This model allows for a vertex $i$ to be highly popular among its cluster (high $\lambda_i^{\text{in}}$), but to be not necessarily popular ($\lambda_i^{\text{out}} = 1$) or even to be very unpopular (small $\lambda_i^{\text{out}}$) with vertices in other clusters.

**Optimal clustering error rate: from edge-density to popularity patterns** An important question to assess the difficulty of the clustering task in a block model is the derivation of the *optimal error rate*. By optimal error rate, we refer to the minimum possible error that the best algorithm achieves when attempting to recover the true cluster assignment of all vertices. This error rate is typically measured in terms of the misclassification rate—that is, the proportion of vertices incorrectly assigned to their true clusters, up to a permutation of the labels. The optimal error rate reflects the information-theoretic limits of the clustering task, because it characterizes how well one could possibly do even with unlimited computational power, given the amount of signal and noise in the data. It also provides a benchmark to evaluate existing algorithms and guides the development of new methods that approach (either theoretically or empirically) these theoretical limits.

Studying the effect of the different model parameters (such as sparsity or degree heterogeneity) on the error rate offers deep insight into the fundamental difficulty of the graph clustering problem across different network settings. Consider a SBM with $k$ clusters of same size $n/k$ and homogeneous interactions as in (1.1). When $1/n \ll p, q \ll 1$, the optimal error rate is asymptotically (Zhang and Zhou, 2016)

$$\exp\left(-\frac{n}{k}\left(\sqrt{p} - \sqrt{q}\right)^2\right).$$

---

[1] A block model is said to have homogeneous interactions if the entries $P_{ij}$ depends only on whether $z_i = z_j$ or $z_i \neq z_j$; otherwise, the model is said to have heterogeneous interactions. Our work focuses on models with heterogeneous interactions, with homogeneous interactions treated as a special case. However, for simplicity, in the Introduction we present results only for the homogeneous setting.

As $p$ and $q$ represent the intra-cluster and inter-cluster edge densities, respectively, the key quantity $(\sqrt{p} - \sqrt{q})^2$ in the expression above captures the influence of edge density: the larger the gap between $p$ and $q$, the easier it is to recover the clusters.

Next, consider a DCBM with $k$ clusters of same size $n/k$ and homogeneous interactions as in (1.2). Under some technical conditions on the degree-correction parameters, Gao et al. (2018) establishes that, when $p, q = o(1)$ with $p/q = O(1)$ and $p = \omega(1/n)$, the optimal error rate is asymptotically

$$\frac{1}{n} \sum_i \exp\left(-\theta_i \frac{n}{k} \left(\sqrt{p} - \sqrt{q}\right)^2\right).$$

Compared to the standard SBM, the difficulty of clustering now varies across vertices and is quantified by the term $\exp\left(-\theta_i n (\sqrt{p} - \sqrt{q})^2 / k\right)$, which depends on each vertex $i \in [n]$ and is monotonically decreasing in $\theta_i$. The optimal error rate corresponds to the average of these quantities over all vertices. This highlights the effect of degree heterogeneity: vertices with larger expected degree are easier to cluster, as their neighborhoods contain more information.

However, the same key quantity $(\sqrt{p} - \sqrt{q})^2$ representing the edge-density signal shows up in the DCBM error rate. Indeed, as mentioned earlier, the degree-correction parameters uniformly inflate or deflate the connection probabilities. As a result, the value of $\theta_i$ impacts the clustering difficulty of vertex $i$ in a predictable and monotonic way. This no longer holds in the PABM, which introduces a richer and more nuanced structure. The first major contribution of this work is to characterize the optimal error rate for clustering under the PABM. As the general expression is somewhat involved, we begin with the simplest case of $k = 2$ clusters of equal size. In this setting, we establish that the optimal error rate is given by

$$\frac{1}{n} \sum_{i \in [n]} \exp\left(-\frac{1}{2} \sum_{j \in [n]} \left(\sqrt{\lambda_i^{\text{in}} \lambda_j^{\text{in}} p} - \sqrt{\lambda_i^{\text{out}} \lambda_j^{\text{out}} q}\right)^2\right).$$

As in the DCBM, the error rate in PABM is expressed as an average over the difficulty of clustering each individual vertex. However, in PABM, these per-vertex difficulties have a more intricate form, and we provide further insight in Sections 2.2 and 2.4. A particularly important observation is the following: suppose $p = q$, so that the expected numbers of intra-cluster and of inter-cluster edges are equal. In this case, the SBM and DCBM reduce to the Erdős-Rényi and Chung–Lu models, respectively, and cluster recovery is fundamentally impossible. Remarkably, this is not true for PABM: cluster recovery may still be possible provided the popularity coefficients $\lambda_i^{\text{in}}$ and $\lambda_i^{\text{out}}$ are different. This reveals a novel aspect of degree heterogeneity captured by PABM but missed by DCBM: *local* differences in intra- and inter-cluster popularity enhance the separability of clusters, even when traditional *global* edge-density signals vanish. Another phenomenon, more subtle, occurs in PABM: the optimal error rate is *not* monotonically increasing when the number of inter-cluster edges increases. We rigorously establish these phenomena in Examples 1 and 2, and illustrate them in our numerical simulations.

**Higher-order eigenvectors for clustering with popularity patterns**  Finally, we perform numerical experiments to evaluate the effectiveness of spectral clustering methods. When the adjacency matrix $A$ is sampled from a block model, it can be decomposed as $A = P + X$, where $P$ is a low-rank matrix encoding the underlying structure, and $X$ is a random noise matrix with zero-mean sub-Gaussian entries. This decomposition forms the basis of spectral methods for graph clustering, where the general approach is to apply a clustering algorithm (such as $k$-means) to a low-dimensional embedding derived from a low-rank approximation of $A$.

In classical models like SBM and DCBM, when $p \neq q$, the rank of $P$ is equal to the number of clusters $k$. However, in PABM, the situation is more complex: the rank of $P$ can be greater than $k$, but cannot be greater than $k^2$. This implies that embeddings based solely on the top-$k$ eigenvectors may miss important structural information. To address this, recent works propose spectral algorithms that incorporate $k^2$ eigenvectors to better capture the richer structure of PABM (Noroozi et al., 2021; Koo et al., 2023). Our numerical experiments demonstrate that these methods outperform traditional spectral approaches that rely only on $k$ eigenvectors, both on synthetic and real datasets.

In the numerical section, we illustrate two surprising results discussed in the theoretical section: the non-monotonic behavior of the error with respect to edge density, and the ability to recover clusters

even when $p = q$. While it would have been possible to use a greedy algorithm to approximate the MLE, we opted for spectral methods because of their widespread use and of their well-established effectiveness for clustering in block models. The experiments demonstrate that the phenomena highlighted in the theoretical section also arise when using spectral algorithms. They show that these behaviors are not merely mathematical artifacts stemming from the increased complexity of PABM relative to DCBM, but that they do occur in practice and are observable in real-world settings.

The paper is structured as follows. We derive the optimal error rate in PABM and provide some examples in Section 2. We present our numerical experiments in Section 3. We discuss some related works in Section 4. Finally, we conclude in Section 5.

**Notations**   $\mathrm{Ber}(p)$, $\mathrm{Exp}(\lambda)$ and $\mathrm{Uni}(a, b)$ denote the Bernoulli distribution with parameter $p$, the exponential distribution with parameter $\lambda$, and the uniform distribution over the interval $[a, b]$. We use the Landau notations $o$ and $O$, and write $f = \omega(g)$ when $g = o(f)$ and $f = \Omega(g)$ when $g = O(f)$.

# 2   Optimal Error Rate in Popularity-Adjusted Block Models

## 2.1   Model Definition and Parameter Space

We consider $n$ vertices partitioned into $k \geq 2$ disjoint blocks. The partition is encoded by a vertex-labeling vector $z^* = (z_1^*, \cdots, z_n^*) \in [k]^n$ so that $z_i^*$ indicates the cluster of vertex $i$. These $n$ vertices interact pairwise, giving rise to undirected edges, and these pairwise interactions are grouped by a symmetric matrix $A \in \{0, 1\}^{n \times n}$ called the adjacency matrix. The *Popularity Adjusted Block Model* supposes that, conditionally on the block structure, the upper-diagonal elements $(A_{ij})_{i>j}$ are independent Bernoulli random variables such that, conditionally on $z_i^*$ and $z_j^*$,

$$A_{ij} \mid z_i^*, z_j^* \sim \mathrm{Ber}\left(\rho_n \lambda_{iz_j^*} \lambda_{jz_i^*} B_{z_i^* z_j^*}\right), \tag{2.1}$$

where $(\lambda_{ia})_{i \in [n], a \in [k]}$ are the popularity parameters and $B \in \mathbb{R}_+^{k \times k}$ is the connectivity matrix across clusters. The parameter $\rho_n$ controls the graph sparsity, as the average degree is of order $n\rho_n$ when the following assumption is made.

**Assumption 1.** *The quantities $B_{ab}$ and $\lambda_{ia}$ are constant (so they do not scale with $n$) for all $i \in [n]$ and $a, b \in [k]$.*

Given a realization of a PABM, we aim to infer the latent block structure $z^*$. Let $\hat{z} = \hat{z}(A)$ be an estimate of $z^*$, and define the clustering error as

$$\mathrm{loss}(z^*, \hat{z}) \;=\; \frac{1}{n} \min_{\tau \in \mathrm{Sym}(k)} \mathrm{Ham}(z^*, \tau \circ \hat{z}), \tag{2.2}$$

where $\mathrm{Sym}(k)$ is the set of permutations of $[k]$ and $\mathrm{Ham}(\cdot, \cdot)$ is the Hamming distance. We are interested in the expected loss of an estimator, namely $\mathbb{E}\left[\mathrm{loss}(z^*, \hat{z}(A))\right]$, where the expectation is taken with respect to the random variable $A$ sampled from (2.1).

## 2.2   A Key Information-Theoretic Divergence

For any $z \in [k]^n$, denote $P_{ij}(z) = \rho_n \lambda_{iz_j} \lambda_{jz_i} B_{z_i z_j}$. To understand the difficulty of correctly clustering a given vertex $i$, we introduce an alternative cluster labeling $\tilde{z}^{ia} \in [k]^n$ such that $\tilde{z}_j^{ia} = z_j^*$ for all $j \neq i$, while $\tilde{z}_i^{ia} = a \in [k] \setminus \{z_i^*\}$. In other words, the cluster labeling $\tilde{z}^{ia}$ agrees with $z^*$ for all vertices except for $i$, which is placed in cluster $a$ instead of being in cluster $z_i^*$. To shorten the notations, let $P^* = P(z^*)$ and $\tilde{P}^{ia} = P(\tilde{z}^{ia})$. The difficulty of correctly recovering the cluster of vertex $i$ depends on how hard it is to statistically distinguish whether the observed graph was generated from the true model $P^*$ or from the alternative model $\tilde{P}^{ia}$. This is a classical hypothesis testing problem: the more similar the distributions induced by $P^*$ and $\tilde{P}^{ia}$, the less distinguishable two graphs drawn from these two models are, and thus the harder it is to infer the correct cluster assignment for vertex $i$. The statistical difficulty of this test is quantified by the Chernoff divergence $\Delta(i, a)$, which measures the exponential rate at which the error probability decays when testing

between these two competing models. More precisely,

$$\Delta(i, a) = \max_{t \in (0,1)} (1 - t) \text{Ren}_t \left( \bigotimes_{j \neq i} \text{Ber} \left( \tilde{P}_{ij}^{ia} \right), \bigotimes_{j \neq i} \text{Ber} \left( P_{ij}^* \right) \right), \quad (2.3)$$

where $\text{Ren}_t$ is the Rényi divergence of order $t$. Moreover, by using the linearity of Rényi divergence with respect to multiplication and the sparsity of the model (that is, $P_{ij} = o(1)$ for all $i, j$), we have

$$\Delta(i, a) = (1 + o(1)) \max_{t \in (0,1)} \sum_{j \neq i} \left( t \tilde{P}_{ij}^{ia} + (1 - t) P_{ij}^* - (\tilde{P}_{ij}^{ia})^t (P_{ij}^*)^{1-t} \right).$$

Among all alternative models $\tilde{P}^{ia}$, the most challenging to distinguish from the true model $P^*$ is the one with the smallest Chernoff divergence $\Delta(i, a)$. We thus define

$$\text{Chernoff}(i, z^*) = \min_{a \neq z_i^*} \Delta(i, a),$$

which captures the hardest hypothesis testing problem associated with recovering the cluster of vertex $i$. Intuitively speaking, the larger the value of $\text{Chernoff}(i, z^*)$, the easier it is to correctly recover $z_i^*$, as all alternative models defined above are sufficiently different from $P^*$. The following assumption asserts that for every $i \in [n]$, the quantity $\text{Chernoff}(i, z^*)$ is unbounded. This assumption is necessary to ensure that the recovery of $z_i^*$ is asymptotically possible.

**Assumption 2.** *Suppose that* $\min_{i \in [n]} \text{Chernoff}(i, z^*) = \omega(1)$.

## 2.3   Main Result: Optimal Error Rate in PABM

For any $z \in [k]^n$, denote by $n_a(z) = \sum_{i \in [n]} \mathbb{1}\{z_i = a\}$ the size of the cluster $a \in [k]$. Let $\pi \in [0, 1]^k$ such that $\sum_a \pi_a = 1$ and define

$$\mathcal{Z}_n(\pi, \epsilon) = \left\{ z \in [k]^n : \frac{n_a(z)}{n} \in [(1 - \epsilon)\pi_a, (1 + \epsilon)\pi_a] \, \forall a \in [k] \right\}.$$

Let $\Lambda = (\lambda_{ia})_{i \in [n], a \in [k]}$ be a matrix with non-negative coefficients such that $\|\Lambda_{\cdot a}\|_1 = n\pi_a$ and $B \in \mathbb{R}_+^{k \times k}$ be a matrix of full rank.

**Theorem 1** (Lower-bound on the clustering error). *Let* $z^* \in \mathcal{Z}_n(\pi, \epsilon)$ *and $A$ being sampled from* (2.1). *Suppose Assumption 2 holds. Then, there exists some $\eta = o(1)$ such that*

$$\inf_{\hat{z}} \mathbb{E} \left[ \text{loss}(z^*, \hat{z}) \right] \geq \frac{(1 - \epsilon) \min_a \pi_a}{4} \left( \frac{1}{n} \sum_{i \in [n]} e^{-\text{Chernoff}(i, z^*)} \right)^{1+\eta},$$

*where the* inf *is taken over all estimators* $\hat{z} = \hat{z}(A)$.

**Theorem 2** (Achievability). *Let* $z^* \in \mathcal{Z}_n(\pi, \epsilon)$ *and $A$ being sampled from* (2.1). *Suppose Assumptions 1 and 2 hold. Then, there exists an estimator $\hat{z}$ such that*

$$\mathbb{E} \left[ \text{loss}(z^*, \hat{z}) \right] \leq \left( \frac{1}{n} \sum_{i \in [n]} e^{-\text{Chernoff}(i, z^*)} \right)^{1+\eta}$$

*for some $\eta = o(1)$*

The gap between the lower bound (Theorem 1) and the achievability (Theorem 2) stems only from second-order terms. Indeed, the sequences $\eta$ appearing in Theorems 1 and 2 are not identical. Moreover, the multiplicative factor $\frac{(1-\epsilon) \min_a \pi_a}{4}$ of constant order can be absorbed into the sequence $\eta$, as the term $\frac{1}{n} \sum_{i \in [n]} e^{-\text{Chernoff}(i, z^*)}$ vanishes as $n \to \infty$. We chose to display this factor explicitly in our bounds so that the sequence $\eta$ does not depend on the parameter $\epsilon$.

We show in Section 2.4 how Theorems 1 and 2 recover known results in SBM and DCBM, and in Section 2.5, how they reveal novel properties that did not exist in previous models, when they are specialized to PABM. Table 1 summarizes all three classes of block models considered in this paper.

| | SBM | DCBM | PABM |
|---|---|---|---|
| Homogeneous | $P_{ij} = \begin{cases} p_0\rho_n & \dots \\ q_0\rho_n & \dots \end{cases}$ | $P_{ij} = \begin{cases} \theta_i\theta_j p_0\rho_n & \dots \\ \theta_i\theta_j q_0\rho_n & \dots \end{cases}$ | $P_{ij} = \begin{cases} \lambda_i^{\text{in}}\lambda_j^{\text{in}} p_0\rho_n & \dots \text{ if } z_i = z_j, \\ \lambda_i^{\text{out}}\lambda_j^{\text{out}} q_0\rho_n & \dots \text{ otherwise.} \end{cases}$ |
| Heterogeneous | $P_{ij} = B_{z_i z_j}\rho_n$ | $P_{ij} = \theta_i\theta_j B_{z_i z_j}\rho_n$ | $P_{ij} = \lambda_{iz_j}\lambda_{jz_i} B_{z_i z_j}\rho_n$ |

Table 1: Different expressions of the elements of the matrix $P$ for the block model variants considered in this paper. The quantity $\rho_n$ is related to the graph sparsity (as the expected degree is of order $n\rho_n$). All other quantities are strictly positive and independent of $n$.

## 2.4 Recovering Known Optimal Error Rates in SBM and DCBM

**Inhomogeneous SBM** Let $\lambda_{ia} = 1$ for all $i \in [n]$ and $a \in [k]$, so that we recover the SBM with inhomogeneous interactions in which $P_{ij} = \rho_n B_{z_i^* z_j^*}$. By linearity of the Rényi divergence, we have

$$
\Delta(i,a) = \max_{t\in(0,1)} (1-t) \sum_{b=1}^{k} n\pi_b \, \text{Ren}_t \left( \text{Ber}\left(\rho_n B_{ab}\right), \text{Ber}\left(\rho_n B_{z_i^* b}\right) \right)
$$

$$
= (1+o(1)) n\rho_n \underbrace{\max_{t\in(0,1)} (1-t) \sum_{b=1}^{k} \pi_b \left( t B_{ab} + (1-t) B_{z_i^* b} - B_{ab}^t B_{z_i^* b}^{1-t} \right)}_{\text{CH}_{AS}(a, z_i^*)}.
$$

The quantity $\text{CH}_{AS}(a,b)$ is called the Chernoff-Hellinger divergence (Abbe and Sandon, 2015). When $n\rho_n = \omega(1)$, we observe that

$$
\frac{1}{n} \sum_{i\in[n]} e^{-n\rho_n \min_{a\in[k]\setminus\{z_i^*\}} \text{CH}_{AS}(a, z_i^*)} = e^{-(1+o(1))n\rho_n \min_{a\neq b\in[k]} \text{CH}_{AS}(a,b)},
$$

and we recover the instance-optimal error rate in SBM with inhomogeneous interactions (Yun and Proutière, 2016). Finally, the Chernoff-Hellinger divergence has a simple expression in the case of homogeneous interactions. Indeed, when $B_{ab} = p_0 \mathbb{1}(a=b) + q_0 \mathbb{1}(a\neq b)$ and the clusters are of equal-size ($\pi_a = 1/k$), the divergence simplifies to $\min_{a\neq b\in[k]} \text{CH}_{AS}(a,b) = \frac{n\rho_n}{k}(\sqrt{p_0} - \sqrt{q_0})^2$.

**Degree-Corrected Block Model** Suppose that $\lambda_{iz_j^*} = \theta_i$, so that the PABM boils down to a DCBM with homogeneous interactions in which $P_{ij} = \theta_i\theta_j B_{z_i^* z_j^*}\rho_n$. For the simplicity of the discussion, we consider cluster of equal-size (*i.e.*, $\pi_a = 1/k$ for all $a \in [k]$), and homogeneous interactions (*i.e.*, $B_{ab} = p_0 \mathbb{1}\{a=b\} + q_0 \mathbb{1}\{a\neq b\}$). Consider a vertex $i$ in a cluster $z_i^*$ and let $a \in [k]\setminus\{z_i^*\}$. We have $\Delta(i,a) = \theta_i \frac{n\rho_n}{k} \left( \sqrt{p_0} - \sqrt{q_0} \right)^2$, where we used $\sum_{i:\, z_i^*=a} \theta_i = 1$. Thus, we recover the asymptotic optimal error-rate $\frac{1}{n} \sum_i e^{-\theta_i \frac{n\rho_n}{k} \left( \sqrt{p_0} - \sqrt{q_0} \right)^2}$ established in Gao et al. (2018).

## 2.5 Optimal Error Rate in Homogeneous PABM

We now show how Theorems 1 and 2, when applied to PABM, reveal novel phenomena. Suppose $B_{ab} = p_0 \mathbb{1}\{a=b\} + q_0 \mathbb{1}\{a\neq b\}$ and $\lambda_{ia} = \lambda_i^{\text{in}} \mathbb{1}\{z_i^* = a\} + \lambda_i^{\text{out}} \mathbb{1}\{z_i^* \neq a\}$. We have

$$
\text{Chernoff}(i, z^*) = \frac{n\rho_n}{2k} \left( \delta_{z_i^*} + \min_{a\neq z_i^*} \delta_a \right) \text{ where } \delta_b = \frac{1}{n/k} \sum_{j:\, z_j^*=b} \left( \sqrt{\lambda_i^{\text{in}}\lambda_j^{\text{in}} p_0} - \sqrt{\lambda_i^{\text{out}}\lambda_j^{\text{out}} q_0} \right)^2.
$$

**Proposition 3.** *Consider a PABM with homogeneous interactions with $p_0 q_0 > 0$, and $k$ equal-size communities. Suppose that $\lambda_1^{\text{out}} = \cdots = \lambda_n^{\text{out}} = 1$ and that the coefficients $\lambda_1^{\text{in}}, \cdots, \lambda_n^{\text{in}}$ are sampled iid from $\text{Uni}(1-c, 1+c)$ with $c \in (0,1)$. Denote $\gamma_c = \frac{1}{3c}\left((1+c)^{3/2} - (1-c)^{3/2}\right)$. We have*

$$
\frac{1}{n} \sum_{i\in[n]} e^{-\text{Chernoff}(i, z^*)} = (1+o(1)) e^{-\frac{n\rho_n}{k} q_0(1-\gamma_c^2)} J_n,
$$

*where*

$$J_n = \frac{k}{2cp_0 n\rho_n} \left( e^{-\frac{n\rho_n}{k} p_0 u_+^2} - e^{-\frac{n\rho_n}{k} p_0 u_-^2} \right) + \frac{\gamma_c}{2c} \sqrt{\frac{k\pi}{n\rho_n q_0}} \left( \mathrm{erf}\left( \frac{n\rho_n}{k} p_0 u_+ \right) - \mathrm{erf}\left( \frac{n\rho_n}{k} p_0 u_- \right) \right),$$

*with $u_\pm = \sqrt{1 \pm c} - \gamma_c \sqrt{q_0/p_0}$ and $\mathrm{erf}(t) = 2/\sqrt{\pi} \int_0^t e^{-t^2} \, \mathrm{d}t$ is the Gauss error function.*

Although the expression of $J_n$ is quite involved, we can give two interesting particular cases. Firstly, to highlight the effect of degree heterogeneity, suppose that $p_0 = q_0$. In this extreme case where we expect the same number of interactions within and across clusters, the only information comes from the degree heterogeneity. Therefore, many existing graph clustering algorithms are expected to fail (as indeed shown in Section 3). However, the quantity $\mathrm{Chernoff}(i, z^*)$ does not vanish. Therefore, even in this extreme setting, consistent recovery is possible, as highlighted in the following example. This stands out in stark contrast to the standard and degree-corrected block models, where setting $p_0 = q_0$ causes the model to collapse into an Erdős-Rényi graph and a Chung-Lu graph, respectively—both of which contain no information about the underlying cluster structure.

**Example 1.** *Consider the setting of Proposition 3 where $p_0 = q_0$. For $c = 0$ the model reduces to an Erdős-Rényi graph with edge-probability $p$, and thus no recovery is possible. However, for $c > 0$,*

$$\frac{1}{n} \sum_{i \in [n]} e^{-\mathrm{Chernoff}(i, z^*)} = (1 + o(1)) e^{-\frac{n\rho_n}{k} p_0 (1 - \gamma_c^2)}.$$

*Observe that $1 - \gamma_c^2$ is strictly positive and increasing in $c \in (0, 1]$. Thus, if $c > 0$, the optimal error rate satisfies $\frac{1}{n} \sum_{i \in [n]} e^{-\mathrm{Chernoff}(i, z^*)} = o(1)$, yielding that cluster recovery is possible. Moreover, this rate is monotonically decreasing in $c$.*

In the following example, we fix $c \in (0, 1)$ and $p_0 > 0$, and we let $q_0$ vary between 0 and $p_0$.

**Example 2.** *Consider the setting of Proposition 3 with $\xi = q_0/p_0 \in (0, 1]$. The quantity $\exp(-\frac{n\rho_n}{k} p_0 \xi (1 - \gamma_c^2)) \times J_n$ is not monotonically increasing in $\xi$, but instead first increases with $\xi$, reaches some maximum value, and then decreases. We illustrate this in Figure 3 in Appendix D.3.*

The intuition behind Example 2 is as follows. As $\xi = 0$ the graph is disconnected and the $k$ largest components are aligned with the $k$ clusters. Hence, the difficulty of clustering is at its lowest and can only increases with $\xi$, as additional edges are inter-cluster edges and act as noise. However, when $\xi$ becomes large enough, the difference between the intra- and inter-connectivity patterns, governed by the $\lambda^{\mathrm{in}}$ and $\lambda^{\mathrm{out}}$, becomes more pronounced. As a result, this provides additional information that can be exploited for clustering (as in Example 1). This leads to a trade-off between the benefit brought by the absence of any inter-cluster edges (for learning from well-separated clusters) and the benefit brought by their presence in large numbers (for learning from popularity patterns). This non-monotonic behavior is specific to PABM and does not occur in DCBM. Finally, this non-monotonicity is *not* an artifact of setting $\lambda_i^{\mathrm{out}} = 1$ and sampling the $\lambda_i^{\mathrm{in}}$ from a uniform distribution. This choice was made because of the difficulty to derive a closed-form expression for the optimal error rate. In Appendix D.3, we show numerically that these phenomena persist under alternative distributions for the coefficients $\lambda^{\mathrm{in}}$ and $\lambda^{\mathrm{out}}$.

## 3 Numerical Experiments

In this section, we numerically evaluate the performance of several existing variants of spectral clustering on both synthetic and real-world datasets.[2] Specifically, we compare the following variants:

- *sbm*: Lloyd's algorithm applied to the embedding formed by the $k$ largest (in magnitude) eigenvectors of the adjacency matrix (see Algorithm 2);

- *dcbm*: Lloyd's algorithm applied to an estimate $\hat{P}$ of the connectivity matrix $P$, constructed using the $k$ largest (in magnitude) eigenvectors of the adjacency matrix (see Algorithm 3);

- *pabm*: subspace clustering applied to the embedding formed by the $k^2$ largest (in magnitude) eigenvectors of the adjacency matrix (see Algorithm 5);

---

[2]Our code is available at `https://github.com/mdreveton/neurips-pabm`.

- *osc*: the spectral clustering variant described in Algorithm 4;

- *sklearn*: Lloyd's algorithm applied to the embedding formed by the $k$ smallest eigenvectors of the graph's normalized Laplacian, corresponding to the implementation available in the *scikit-learn* library (see Algorithm 1).

The *sbm* and *dcbm* variants are tailored for graphs generated from SBM and DCBM, respectively, and are known to recover clusters accurately under these models (Zhang, 2024; Gao et al., 2018). In contrast, PABM exhibits a more complex structure, as the rank of the matrix $P$ can exceed $k$, but cannot be greater than $k^2$. We refer to (Koo et al., 2023; Noroozi et al., 2021) and to Section E.4 of the Appendix for examples. To accommodate this higher-rank structure, the *pabm* and *osc* variants rely on an embedding based on $k^2$ eigenvectors rather than the traditional $k$, allowing them to capture the higher-rank structure of PABM more effectively.

In the finalization phase of the manuscript, we became aware of two more algorithms designed for community recovery in PABM, namely Thresholded Cosine Spectral Clustering (*tcsc*) and a Greedy Subspace Projection Clustering (*gspc*), introduced in Yuan et al. (2025) and in Bhadra et al. (2025), respectively. To avoid overburdening this section, we refer the interested reader to the Appendix E.3 for a description of these algorithms. We also provide in the Appendix E the pseudo-code of all the algorithms.

In all experiments, we report the accuracy, defined as one minus the loss in (2.2). It is equal to the proportion of correctly clustered vertices.

## 3.1 Synthetic Data Sets

We first consider homogeneous PABM whose interaction probabilities are given by

$$P_{ij} = \begin{cases} \lambda_i^{\text{in}} \lambda_j^{\text{in}} \rho & \text{if } z_i^* = z_j^*, \\ \lambda_i^{\text{out}} \lambda_j^{\text{out}} \xi \rho & \text{otherwise.} \end{cases} \tag{3.1}$$

The parameter $\rho \in (0, 1)$ controls the overall sparsity of the network, while the parameter $\xi \in [0, 1]$ controls the fraction of edges across clusters (in particular, $\xi = 0$ implies no inter-cluster edges while $\xi = 1$ implies the same expected number of edges between any pair of clusters). As in Examples 1 and 2, we let $\lambda_i^{\text{out}} = 1$ and sample the coefficients $\lambda^{\text{in}}$ from the uniform distribution in $(1 - c, 1 + c)$.

In Figure 1a, we let $\xi = 1$ and vary $c$. This is precisely the setting of Example 1. We observe that *pabm* and *osc* variants, which are specifically designed for PABM, recover the clusters when $c$ is large enough, whereas the variants tailored for SBM and DCBM fail to do so. This illustrates that *pabm* and *osc* successfully learn the clusters without edge-density signal by using the difference in the individual vertex degree connectivity patterns. In Figure 1b, we set $c = 0.8$ and let $\xi$ vary. We observe that the acuracy of *pabm* and *osc* is *not* monotonically decreasing with $\xi$. In fact, it goes to a minimum value before increasing again. This illustrates the phenomenon described in Example 2. In contrast, the accuracy obtained by *sbm* and *dcbm* variants monotonically decreases, because increasing $\xi$ from 0 to 1 monotonically decreases the edge-density signal.[3]

To further highlight the impact of the embedding dimension on the clustering accuracy, we plot in Figure 2 the accuracy of the different spectral clustering variant as a function of embedding dimension $d$. We observe that the performance of pabm and osc improves significantly as the dimension increases from $d = 3$ to $d = 6$, after which it reaches a plateau. In contrast, the performance of the sbm and dcbm variants remains unchanged with increasing $d$.

## 3.2 Real Data Sets

In this section, we show on real datasets that spectral algorithms that use more eigenvectors such as *pabm* and *osc* outperform the traditional variants that use only $k$ eigenvectors. Table 3 in Appendix F.3 summarizes some statistics of the dataset used. Table 2 shows the accuracy obtained by the different variants of spectral clustering on the real data sets.

---

[3]In both cases, the accuracy achieved by *sklearn* matches that of *dcbm* and is omitted from the figures.

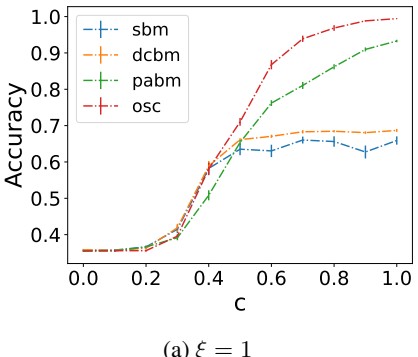

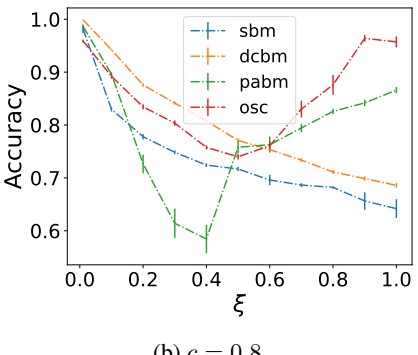

(a) $\xi = 1$

(b) $c = 0.8$

Figure 1: Performance of graph clustering on homogeneous PABM, where the matrix $P$ is given in Equation (3.1). We sampled graphs with $n = 900$ vertices in $k = 3$ clusters of same size, average edge density $\rho = 0.05$. In both figures, the $\lambda_i^{\text{in}}$ are iid sampled from $\text{Uni}(1 - c, 1 + c)$ and $\lambda_i^{\text{out}} = 1$ for all $i$. Accuracy is averaged over 15 realizations, and error bars show the standard errors.

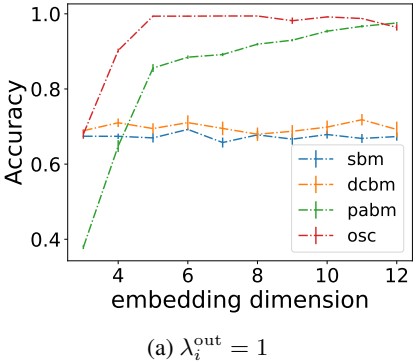

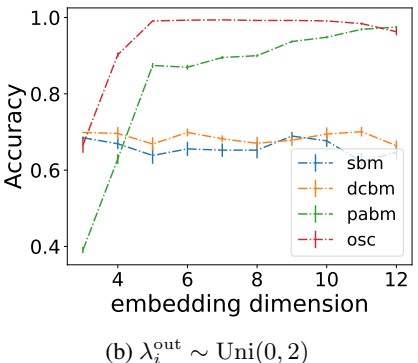

(a) $\lambda_i^{\text{out}} = 1$

(b) $\lambda_i^{\text{out}} \sim \text{Uni}(0, 2)$

Figure 2: Effect of the embedding dimension on the performance of graph clustering on homogeneous PABM, where the matrix $P$ is given in Equation (3.1). We sampled graphs with $n = 900$ vertices in $k = 3$ clusters of same size, average edge density $\rho = 0.05$. In both figures, the $\lambda_i^{\text{in}}$ are iid sampled from $\text{Uni}(0, 2)$. Accuracy is averaged over 15 realizations, and error bars show the standard errors.

|  | sbm | dcbm | pabm | osc | tcsc | gspc | sklearn |
|---|---|---|---|---|---|---|---|
| politicalBlogs | 0.63 | **0.95** | 0.91 | **0.95** | 0.65 | **0.95** | 0.52 |
| liveJournal-top2 | 0.56 | 0.61 | **0.99** | 0.59 | 0.98 | 0.60 | **0.99** |
| citeseer | 0.27 | 0.38 | 0.45 | 0.56 | 0.33 | 0.51 | **0.58** |
| cora | 0.34 | 0.37 | **0.47** | **0.47** | 0.30 | 0.42 | 0.27 |
| mnist | 0.44 | 0.54 | **0.88** | 0.74 | 0.11 | 0.79 | 0.78 |
| fashionmnist | 0.22 | 0.41 | **0.63** | 0.61 | 0.60 | 0.50 | 0.60 |
| cifar10 | 0.17 | 0.43 | **0.74** | 0.58 | 0.49 | 0.62 | 0.71 |

Table 2: Accuracy of several spectral clustering variants on real data sets.

## 4    Related Work

**Optimal clustering error rate.**    A rich line of research focused on characterizing the optimal error rates for clustering in stochastic block models and their variants. Early results established the minimax error rate in the SBM (Zhang and Zhou, 2016), while later work extended these insights to more general models such as the degree-corrected block model (Gao et al., 2018). Further developments have addressed more complex network structures, such as categorical edge types (Yun and Proutière, 2016), weighted interactions (Xu et al., 2020), and more general interaction patterns (Avrachenkov et al., 2022). These studies leverage information-theoretic tools to derive minimax bounds and to uncover the fundamental limits of clustering error. Parallel developments have taken place in the

mixture model literature, where optimal error rates have been studied extensively, particularly in Gaussian mixture models (Lu and Zhou, 2016; Cai et al., 2019; Chen and Zhang, 2024) and in more general mixture models (Dreveton et al., 2024). In both settings, a central objective is to understand how the separation between components governs the intrinsic difficulty of the clustering task.

**Clustering with higher-order eigenvectors.** Several studies have identified the benefits of incorporating higher-order eigenvectors beyond the first $k$ in spectral graph clustering. In networks whose connections depend on both cluster membership and spatial position, Avrachenkov et al. (2021) demonstrated that the second eigenvector of the graph Laplacian typically aligns with the geometric structure rather than with the cluster structure. As a result, traditional spectral methods that rely solely on the leading eigenvectors often produce geometric partitioning that fails to accurately capture the underlying cluster structure. Their analysis reveals that incorporating additional eigenvectors beyond the conventional first $k$ can provide crucial information for distinguishing between geometric proximity and actual cluster membership.

In sparse networks with strong degree heterogeneity—where some vertices have significantly higher degree than others—spectral clustering based on the top $k$ eigenvectors of the adjacency matrix often fails. In such cases, the leading eigenvectors tend to localize around high-degree vertices, rather than capturing the underlying cluster structure. Trimming-based approaches have been proposed to mitigate this issue by down-weighting or removing influential high-degree vertices (Le et al., 2017). Alternatively, using the normalized Laplacian shifts the problem: its leading eigenvectors may become concentrated on peripheral substructures, such as dangling trees, while the cluster signal may still lie in higher-order eigenvectors. To address this, regularization techniques have been introduced to stabilize the spectral embedding and improve clustering performance (Qin and Rohe, 2013).

Although the previous paragraphs illustrate two different settings where higher-order eigenvectors are crucial for uncovering cluster structure, they also share a key limitation: the leading eigenvectors are largely uninformative, and only the higher-order ones carry meaningful clustering information. PABM is fundamentally different, as potentially all $k^2$ eigenvectors can be informative for clustering. This richer spectral structure opens new avenues for designing more effective spectral algorithms.

## 5   Conclusion

We established the optimal error rate for clustering under the PABM, providing a precise information-theoretic characterization of the fundamental limits of clustering in this rich and flexible model. Our results highlight how heterogeneity in vertex popularity fundamentally alters the clustering landscape, and how this is reflected in the spectral structure of the network. While our analysis provides a solid theoretical foundation, several important questions remain open. A deeper theoretical understanding of practical algorithms such as OSC and subspace clustering remains a key challenge. Another important direction for future work is model selection: developing principled methods to distinguish between models such as DCBM and PABM, and to infer key parameters like the number $k$ of clusters or the rank of the connection probability matrix $P$. Addressing these challenges is essential to translate theoretical insights into robust, data-driven tools for network analysis.

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

# A  Additional Discussion of Theoretical Results

## A.1  Instance-Optimal versus Minimax Setting

In our study of optimal clustering rates in PABM, we did not made any assumption on the matrix $B$ (beyond being symmetric). As a result, our analysis derives the optimal error rate for a specific instance of PABM (similar to the instance-wise analysis in Yun and Proutière (2016) for the edge-labeled SBM) rather than a minimax error rate (as in Zhang and Zhou (2016) for SBM and Gao et al. (2018) for DCBM). Both approaches are valuable: the minimax framework requires defining a parameter space to which $B$ belongs (typically the space of matrices having diagonal values larger than or equal to $p$ and off-diagonal values smaller than or equal to $q$, but offers no guarantees when the matrix $B$ lies outside this space, while the instance optimal-rate restricts to a specific but arbitrary matrix $B$.

Moreover, we wish to emphasize an important point: a rate-optimal algorithm in the minimax setting may not be rate-optimal for specific instances, even when those instances fall within the defined parameter space. For example, Lloyd's algorithm is minimax-optimal over the class of sub-Gaussian mixture models Lu and Zhou (2016), but it fails to be instance-optimal for Gaussian mixture models with anisotropic covariance structures Chen and Zhang (2024).

For the parameter space described two paragraphs above (matrix $B$ with diagonal values larger than or equal to $p$ and off-diagonal values smaller than or equal to $q$), the worst-case rate for SBM and DCBM arises when $B_{aa} = p$ for all $a \in [k]$ and $B_{ab} = q$ for all $a \neq b$, leading to a minimax rate involving the term $(\sqrt{p} - \sqrt{q})^2$. In contrast, for PABM, the situation is more complex because of the additional dependence on the individual parameters $\lambda_{ia}$. As a result, we do not believe that a simple closed-form expression for the minimax rate in PABM is attainable. Indeed, as shown in Example 2, reducing the gap between $p$ and $q$ does not necessarily increases the optimal error-rate.

## A.2  Overview of the Proofs

The overall structure of the proofs for Theorems 1 and 2, which establish the optimal error rate, is similar to that used for SBM and DCBM (in Zhang and Zhou (2016) and Gao et al. (2018), respectively). However, the PABM setting introduces additional technical complexity that requires a more refined analysis.

(i) For the lower-bound, a first challenge is to address the minimum over all permutations in the definition of the error loss. Hence, rather than directly examining $\inf_{\hat{z} \in [k]^n} \mathbb{E}[n^{-1}\text{loss}(z^*, \hat{z})]$, we follow previous works such as Zhang and Zhou (2016); Gao et al. (2018) and focus on a sub-problem $\inf_{\hat{z} \in \mathcal{Z}} \mathbb{E}[\text{loss}(z^*, \hat{z})]$, where $\mathcal{Z} \subset [k]^n$ is chosen such that $\text{loss}(z^*, \hat{z}) = \text{Ham}(z^*, \hat{z})/n$ for all $z^*, \hat{z} \in \mathcal{Z}$. This sub-problem is simple enough to analyze, while still capturing the hardness of the original clustering problem. Next, we use a result from Dreveton et al. (2024) to show that the Bayes risk $\inf_{\hat{z}_i} \mathbb{P}(\hat{z}_i \neq z_i^*)$ for the misclustering of a single vertex $i$ is asymptotically $e^{-(1+o(1))\text{Chernoff}(i, z^*)}$.

More precisely, (Dreveton et al., 2024, Lemma 2) establishes the worst-case error rate for a binary hypothesis testing problem where the observed random variable is drawn from either distribution $f_1$ or $f_2$ (corresponding to hypothesis $H_1$ and $H_2$, respectively). Both $f_1$ and $f_2$ are arbitrary and known probability density functions. By the Neyman–Pearson lemma, the likelihood ratio test (equivalent to the MLE in this context) minimizes the probability of error, thereby ruling out all other estimators for this problem. The error of the MLE is then upper-bounded using Chernoff's method and lower-bounded using a large deviation argument. In our setting, the hypothesis is formulated in Equation (B.1), where $f_1$ and $f_2$ are product distributions of Bernoulli random variables.

(ii) The proof of the achievability is however more involved, and required a new approach. It begins, similarly to prior work on block models, by upper-bounding $\mathbb{E}[\text{loss}(z^*, \hat{z})]$ (where $\hat{z}$ is the MLE) by $\sum_m \mathbb{P}(\text{Ham}(z^*, \hat{z}) = m)$. Thus, the core difficulty relies in upper-bound the quantities $\mathbb{P}(L(z) > L(z^*))$ for any $z$ such that $\text{Ham}(z^*, z) = m$ (where $L(z)$ denotes the likelihood of $z$ given an observation of $A$). This is more challenging in PABM than in SBM and DCBM. Indeed, unlike in SBM (and to some extent DCBM), the likelihood ratio $L(z)/L(z^*)$ cannot be easily simplified. As for SBM and DCBM, we rely on Chernoff bounds to obtain $\mathbb{P}(L(z) > L(z^*)) \leq \mathbb{E}[e^{t \log(L(z)/L(z^*))}]$ for any $t > 0$. But, in SBM and DCBM, one can use $t = 1/2$ and obtain clean exponential bounds

whose terms are $\exp(-\theta_u\theta_v(\sqrt{p} - \sqrt{q})^2)$. For PABM, the optimal $t$ to use depends intricately on the misclassified set $\{u\colon z_u \neq z_u^*\}$, and thus on $z$ itself. To address this, we adopt a more refined approach: we decompose the upper bound into three components $T_1(t)$, $T_2(t)$, and $T_3(t)$, and select a tailored value of $t$ for each labeling $z$. This additional complexity distinguishes our analysis from earlier work and reflects the greater structural richness of PABM compared to SBM and DCBM.

### A.3 Extension when Assumption 2 Fails

The situation when Assumption 2 fails is slightly delicate. Suppose firstly that Assumption 2 fails such that $\max_i \mathrm{Chernoff}(i, z^*) = O(1)$. In that case, the optimal clustering error cannot vanish. Indeed, using arguments similar to those in Zhang and Zhou (2016), we can establish that the optimal error rate is lower-bounded by a non-zero constant $c > 0$. More generally, we introduce the set $S = \{i \in [n]\colon \mathrm{Chernoff}(i, z^*) = O(1)\}$ of vertices having a non-vanishing error of being misclustered. Assumption 2 fails whenever $S \neq \emptyset$. By refining the proof of Theorem 1, we can obtain a lower-bound for the clustering error of any algorithm of the form:

$$\inf_{\hat{z}} \mathbb{E}\left[\mathrm{loss}(z^*, \hat{z})\right] \;\geq\; \frac{(1-\epsilon)\min_a \pi_a}{4}\left(\frac{1}{|S^c|}\sum_{i \in S^c} e^{-\mathrm{Chernoff}(i,z^*)} + c\frac{|S|}{n}\right)^{1+\eta}.$$

This decomposition reflects that a constant fraction of nodes (those in $S$) are intrinsically hard to classify, while the rest exhibit standard exponential error decay. When Assumption 2 holds, $S = \emptyset$ and the lower bound matches the result in Theorem 1. (And observe that the case $\max_i \mathrm{Chernoff}(i, z^*) = O(1)$ discussed earlier is equivalent to $|S| = n$, and we recover $\inf_{\hat{z}} \mathbb{E}\left[\mathrm{loss}(z^*, \hat{z})\right] \geq c$ for some non-vanishing constant $c > 0$.)

Showing that the MLE attains this bound when $S \neq \emptyset$ appears plausible but requires additional technical work.

## B  Proof of Theorem 1

### B.1  Clustering one Vertex at a Time: the Genie-aided Problem

Let $i \in [n]$ and suppose a genie gives you $z_{-i}^*$, *i.e.,* the community labels of all nodes but $i$. Denote $H_a^{(i)}\colon z_i^* = a$ the hypothesis that node $i$ belongs to the cluster $a \in [k]$. Letting $X = A_{i\cdot}$ being the $i$-th row of the adjacency matrix, the hypothesis testing resumes to

$$H_a^{(i)} : \; X \;\sim\; \bigotimes_{j \neq i} \mathrm{Ber}\left(\rho_n \lambda_{iz_j^*}\lambda_{ja} B_{z_j^* a}\right). \tag{B.1}$$

The worst-case error of a testing procedure $\phi\colon \{0,1\}^{n-1} \to \{1, \cdots, k\}$ is

$$r(\phi) \;=\; \max_{a \neq b} \mathbb{P}\left(\phi(X) = a \mid H_b\right).$$

By the Neyman-Pearson lemma, we have $\phi^{\mathrm{MLE}} = \arg\min_\phi r(\phi)$ where

$$\phi^{\mathrm{MLE}}(A_{i\cdot}) \;=\; \arg\max_{a \in [k]} \prod_{j \neq i}\left(1 - \rho_n \lambda_{iz_j^*}\lambda_{ja} B_{z_j^* a}\right)^{1-A_{ij}}\left(\rho_n \lambda_{iz_j^*}\lambda_{ja} B_{z_j^* a}\right)^{A_{ij}}.$$

Recall that the quantity $\Delta_{ia}(z^*, \Lambda)$ is defined in (2.3) by

$$\Delta(i, a) \;=\; \sup_{t \in (0,1)} (1-t)\mathrm{Ren}_t\left(\bigotimes_{j \neq i}\mathrm{Ber}\left(\rho_n \lambda_{iz_j^*}\lambda_{ja}\right), \bigotimes_{j \neq i}\mathrm{Ber}\left(\rho_n \lambda_{iz_j^*}\lambda_{jz_i^*}\right)\right).$$

(Dreveton et al., 2024, Lemma 2) shows that for all $a \neq z_i^*$ we have

$$\mathbb{P}\left(\phi^{\mathrm{MLE}}(A_{i\cdot}) = a \mid z_i^*\right) \;=\; e^{-(1+o(1))\Delta(i,a)},$$

provided that $\Delta(i,a) = \omega(1)$. Furthermore, if $\Delta(i,a) = \omega(\log k)$, union bounds imply that

$$\mathbb{P}\left(\phi^{\mathrm{MLE}}\left(A_{i\cdot}\right) \neq z_i^*\right) = e^{-(1+o(1))\mathrm{Chernoff}(i,a)} \tag{B.2}$$

where

$$\mathrm{Chernoff}(i,z^*) = \min_{a \neq z_i^*} \Delta(i,a)$$

is the Chernoff information associated with this hypothesis testing problem.

## B.2 Lower-bounding the Optimal Error

*Proof of Theorem 1.* For simplicity, we shorten $\mathcal{Z}_n(\pi,\epsilon)$ by $\mathcal{Z}$. Let $z^* \in \mathcal{Z}$ be the true cluster membership vector. We denote the set of vertices in cluster $a$ by $\Gamma_a(z^*) = \{i \in [n] : z_i^* = a\}$. Following the same proof strategy as previous works on clustering block models (Gao et al., 2018; Dreveton et al., 2024), we define a clustering problem over a subset of $[k]^n$ to avoid the issues of label permutations in the definition of the loss function (2.2). For every cluster $a \in [k]$, we define the set $T_a$ of the $|\Gamma_a(z^*)| - \frac{n(1-\epsilon)\pi_{\min}}{4k}$ vertices belonging to cluster $a$ and having the largest $\mathrm{Chernoff}(i,z^*)$. We motivate this as follows. A vertex $i$ with a large $\mathrm{Chernoff}(i,z^*)$ implies that if a genie provides $z_{-i}^*$ (the community labels of all vertices but $i$), the inference of $z_i^*$ is easy. Hence, the set $T_a$ contains the vertices belonging to the cluster $a$ that are the easiest to cluster, and therefore a good estimator $\hat{z}$ should correctly infer the vertices belonging to $T_a$. In contrast, vertices with small $\mathrm{Chernoff}(i,z^*)$ may be impossible to cluster, even with the best estimator, and these are the vertices that matter in deriving the lower-bound. Let $T = \cup_{a \in [k]} T_a$ and define a new parameter space $\tilde{\mathcal{Z}} \subseteq \mathcal{Z}$

$$\tilde{\mathcal{Z}} = \{z : z_i = z_i^* \text{ for all } i \in T \quad \text{and} \quad \frac{|\Gamma_a(z)|}{n} \in [(1-\epsilon)\pi_a, (1+\epsilon)\pi_a]\}.$$

This new space $\tilde{\mathcal{Z}}$ is composed of all vectors $z \in \mathcal{Z}$ that only differ from $z^*$ on the indices $i$'s that do not belong to $T$. By definition of $T$, these vertices are the hardest to cluster. By construction of $\tilde{\mathcal{Z}}$, we have for any $z, z' \in \tilde{\mathcal{Z}}$

$$\mathrm{Ham}(z,z') = \sum_{i=1}^n \mathbb{1}\{z_i \neq z_i'\} \leq |T^c| = k\frac{n(1-\epsilon)\pi_{\min}}{4k}.$$

Because $z \in \tilde{\mathcal{Z}} \subset \mathcal{Z}$, we have by definition of $\mathcal{Z}$ that $\min_{a \in [k]} |\Gamma_a(z)| \geq (1-\epsilon)n\pi_{\min}$. Therefore, the previous inequality ensures that $\mathrm{Ham}(z,z') < 2^{-1} \min_{a \in [k]} |\Gamma_a(z)|$ for all $z, z' \in \mathcal{Z}$. We can thus apply Lemma 4 to establish that

$$\forall z, z' \in \tilde{\mathcal{Z}}: \quad \mathrm{loss}(z,z') = \frac{1}{n}\mathrm{Ham}(z,z') = \frac{1}{n}\sum_{i \in T^c} \mathbb{1}\{z_i \neq z_i'\}. \tag{B.3}$$

For any estimator $\hat{z}$, we can build an estimator $\hat{z}' \in \tilde{\mathcal{Z}}$ such that

$$\hat{z}_i' = \begin{cases} z_i^* & \text{if } i \in T, \\ \hat{z}_i & \text{otherwise,} \end{cases}$$

and this estimator satisfies $\mathrm{loss}(z^*, \hat{z}') \leq \mathrm{loss}(z^*, \hat{z})$. Therefore,

$$\inf_{\hat{z} \in \mathcal{Z}} \mathbb{E}\,\mathrm{loss}(z^*, \hat{z}) \geq \inf_{\hat{z}' \in \tilde{\mathcal{Z}}} \mathbb{E}\,\mathrm{loss}(z^*, \hat{z}') = \frac{1}{n} \inf_{\hat{z}' \in \tilde{\mathcal{Z}}} \mathbb{E}\,\mathrm{Ham}(z^*, \hat{z}),$$

where the last equality follows from (B.3). Hence, we obtain

$$\inf_{\hat{z} \in \mathcal{Z}} \mathbb{E}\,\mathrm{loss}(z^*, \hat{z}) \geq \frac{1}{n} \inf_{\hat{z}} \sum_{i \in T^c} \mathbb{P}\left(\hat{z}_i \neq z_i^*\right) \geq \frac{1}{n} \sum_{i \in T^c} \inf_{\hat{z}_i} \mathbb{P}\left(\hat{z}_i \neq z_i^*\right).$$

From Equation (B.2), we have

$$\inf_{\hat{z}_i} \mathbb{P}\left(\hat{z}_i \neq z_i^*\right) \geq e^{-(1+\eta_i)\mathrm{Chernoff}(i,z^*)},$$

for some $\eta_i = o(1)$. Let $\eta = \max_i \eta_i$. We obtain

$$
\begin{aligned}
\inf_{\hat{z} \in \mathcal{Z}} \mathbb{E} \operatorname{loss}(z^*, \hat{z}) &\geq \frac{|T^c|}{n} \frac{1}{|T^c|} \sum_{i \in T^c} e^{-(1+\eta)\operatorname{Chernoff}(i, z^*)} \\
&\geq \frac{|T^c|}{n} \frac{1}{n} \sum_{i \in [n]} e^{-(1+\eta)\operatorname{Chernoff}(i, z^*)} \\
&= \frac{(1-\epsilon)\pi_{\min}}{4} \frac{1}{n} \sum_{i \in [n]} e^{-(1+\eta)\operatorname{Chernoff}(i, z^*)},
\end{aligned}
$$

where the second inequality uses the fact that $T^c$ collects the indices of the vertices with the smallest $\operatorname{Chernoff}(i, z^*)$, and the last line uses $\frac{|T^c|}{n} = \frac{\alpha(1-\epsilon)\pi_{\min}}{4}$ (by definition of $T$).

Finally, note that we can always chose $\eta$ to be nonnegative and thus the function $x \mapsto x^{1+\eta}$ is convex. Hence, by Jensen's inequality, we have

$$
\frac{1}{n} \sum_{i \in [n]} \left( e^{-\operatorname{Chernoff}(i, z^*)} \right)^{1+\eta} \geq \left( \frac{1}{n} \sum_{i \in [n]} e^{-\operatorname{Chernoff}(i, z^*)} \right)^{1+\eta}.
$$

$\square$

### B.3 Additional Lemma

**Lemma 4** (Lemma C.5 in Avrachenkov et al. (2022)). *Let $z_1, z_2 \in [k]^n$ such that $\operatorname{Ham}(z_1, \tau^* \circ z_2) < \frac{1}{2} \min_{a \in [k]} |\Gamma_a(z_1)|$ for some $\tau^* \in \operatorname{Sym}(k)$. Then $\tau^*$ is the unique minimizer of $\tau \in \operatorname{Sym}(k) \mapsto \operatorname{Ham}(z_1, \tau \circ z_2)$.*

## C Proof of Theorem 2

**Warm-up: notations and MLE** Let $z \in [k]^n$ be any vertex labeling. We denote $L(z) = \mathbb{P}(A \mid z)$ the likelihood of $z$ given the observation $A$. We study the performance of the maximum likelihood estimator $\hat{z} = \hat{z}(A)$ defined by

$$
\hat{z} = \arg\max_{z \in [k]^n} L(z),
$$

where ties are broken arbitrarily. Hence, by definition, the MLE is any estimator $\hat{z}$ such that

$$
L(\hat{z}) \geq L(z) \quad \text{for all } z \in [k]^n.
$$

Moreover, we have

$$
n \mathbb{E}\left[\operatorname{loss}(z^*, \hat{z})\right] = \mathbb{E}\left[ \min_{\sigma \in \operatorname{Sym}([k])} \operatorname{Ham}(z^*, \sigma \circ z) \right] \leq \mathbb{E}\left[\operatorname{Ham}(z^*, z)\right].
$$

We also recall (see (Dreveton et al., 2023, Lemma 7)) that, for any $z, z' \in [k]^n$ we have

$$
\operatorname{loss}(z, z') \leq n(1 - 1/k).
$$

Therefore,

$$
n\mathbb{E}\left[\operatorname{loss}(z^*, z)\right] \leq \sum_{m=1}^{n(1-1/k)} m\mathbb{P}\left(\operatorname{Ham}(z^*, \hat{z}) = m\right).
$$

For technical reasons that will become clear in the end of the proof, we first need to split the sum into two parts. Let $m_0 \geq 1$, whose value will be determined later. We have

$$
\mathbb{E}\left[\mathrm{Ham}(z^*, z)\right] = \sum_{m=1}^{m_0} m \mathbb{P}\left(\mathrm{Ham}(z^*, \hat{z}) = m\right) + \sum_{m=m_0+1}^{n(1-1/k)} m \mathbb{P}\left(\mathrm{Ham}(z^*, \hat{z}) = m\right)
$$

$$
\leq m_0 + \sum_{m=m_0+1}^{n(1-1/k)} m \mathbb{P}\left(\mathrm{Ham}(z^*, \hat{z}) = m\right).
$$

Let us denote $\mathcal{Z}_m$ the set of vertex labeling $z \in [k]^n$ such that $\mathrm{Ham}(z^*, z) = m$. By definition of the maximum likelihood and by union bounds, we have

$$
\mathbb{P}\left(\hat{z} \in \mathcal{Z}_m\right) \leq \mathbb{P}\left(\exists z \in \mathcal{Z}_m \colon L(z) \geq L(z^*)\right) \leq \sum_{z \in \mathcal{Z}_m} \mathbb{P}\left(L(z) \geq L(z^*)\right).
$$

Hence, by combining the previous inequalities, we obtain

$$
\mathbb{E}\left[\mathrm{loss}(z^*, \hat{z})\right] \leq \frac{1}{n}\left(m_0 + \sum_{m=m_0+1}^{n(1-1/k)} m \sum_{z \in \mathcal{Z}_m} \mathbb{P}\left(L(z) \geq L(z^*)\right)\right). \tag{C.1}
$$

A large part of the rest of the proof is devoted to upper-bound $\sum_{z \in \mathcal{Z}_m} \mathbb{P}\left(L(z) \geq L(z^*)\right)$ for an arbitrary $m$. We first observe that

$$
L(z) = \prod_{i<j} \mathbb{P}\left(A_{ij} \mid z_i, z_j\right)
$$

$$
= \prod_{i<j} \left(\rho_n \lambda_{iz_j} \lambda_{jz_i} B_{z_i z_j}\right)^{A_{ij}} \left(1 - \rho_n \lambda_{iz_j} \lambda_{jz_i} B_{z_i z_j}\right)^{1-A_{ij}}.
$$

In all the following, to avoid overburdening the notations, we denote $P_{ij}^z = \rho_n \lambda_{iz_j} \lambda_{jz_i} B_{z_i z_j}$. We also introduce

$$
\Gamma(z, z^*) = \left\{(i,j) \colon 1 \leq i \neq j \leq n \text{ and } (z_i, z_j) \neq (z_i^*, z_j^*)\right\}
$$

We have

$$
\frac{L(z)}{L(z^*)} = \prod_{\substack{i<j \\ (i,j)\in\Gamma(z,z^*)}} \left(\frac{P_{ij}^z}{P_{ij}^*}\right)^{A_{ij}} \left(\frac{1 - P_{ij}^z}{1 - P_{ij}^*}\right)^{1-A_{ij}}.
$$

Therefore, by Chernoff bounds, we have for any $t > 0$,

$$
\mathbb{P}\left(L(z) > L(z^*)\right) = \mathbb{P}\left(e^{t \log \frac{L(z)}{L(z^*)}} > 1\right)
$$

$$
\leq \prod_{\substack{i<j \\ (i,j)\in\Gamma(z,z^*)}} \mathbb{E}\left[e^{t\left(A_{ij} \log \frac{P_{ij}^z}{P_{ij}^*} + (1-A_{ij}) \log \frac{1-P_{ij}^z}{1-P_{ij}^*}\right)}\right]
$$

$$
= \prod_{\substack{i<j \\ (i,j)\in\Gamma(z,z^*)}} e^{-(1-t)\mathrm{Ren}_t\left(P_{ij}^z, P_{ij}^*\right)}. \tag{C.2}
$$

For ease of the exposition, we start by deriving an upper bound on $\sum_{z \in \mathcal{Z}_m} \mathbb{P}\left(L(z) \geq L(z^*)\right)$ in the simplest case $m = 1$. We do the general case $m \geq 1$ later.

**(i) Case $m = 1$.**  Observe that

$$
\mathcal{Z}_1 = \left\{z \in [k]^n \colon \mathrm{Ham}(z, z^*) = 1\right\} = \left\{\tilde{z}^{ua}, u \in [n], a \in [k] \setminus \{z_u^*\}\right\},
$$

where $\tilde{z}_v^{ua} = z_v^*$ for all $u \neq v$ and $\tilde{z}_u^{ua} = a$. Hence,

$$\mathbb{P}\left(\text{Ham}(z^*, \hat{z}) = 1\right) = \mathbb{P}\left(\exists u \in [n], \exists a \in [k] \setminus \{z_u^*\}: L(\tilde{z}^{ua}) > L(z^*)\right)$$

$$\leq \sum_{u=1}^{n} \sum_{a \in [k] \setminus \{z_u^*\}} \mathbb{P}\left(L(\tilde{z}^{ua}) > L(z^*)\right). \tag{C.3}$$

Moreover, for any $u \in [n]$ and $a \in [k] \setminus \{z_u^*\}$, we have

$$\mathbb{P}\left(L(\tilde{z}^{ua}) > L(z^*)\right) \leq e^{-(1-t)\sum_{j \neq u} \text{Ren}_t\left(P_{uj}^{\tilde{z}^{ua}}, P_{uj}^*\right)}.$$

This last inequality is valid for any $t > 0$. Applying it with $t^* = \arg\max_{t \in (0,1)}(1 - t)\sum_{j \neq u} \text{Ren}_t\left(P_{uj}^{\tilde{z}^u}, P_{uj}^*\right)$, we obtain

$$\mathbb{P}\left(L(\tilde{z}^{ua}) > L(z^*)\right) \leq e^{-\Delta_{ua}} \leq e^{-\text{Chernoff}(u,z^*)},$$

because $\text{Chernoff}(u, z^*) = \min_{a \neq z_u^*} \Delta_{ua}$. Hence, using (C.3) we have

$$\mathbb{P}\left(\text{Ham}(z^*, \hat{z}) = 1\right) \leq k\sum_{u=1}^{n} e^{-\text{Chernoff}(u,z^*)}.$$

**(ii) Case $m \geq 2$** Consider now $z$ such that $\text{Ham}(z, z^*) = m$. Introduce $u_1, \cdots, u_m$ the $m \geq 2$ vertices satisfying $z_{u_p} \neq z_{u_p}^*$ for all $p \in [m]$. By definition, for any $v \notin \{u_1, \cdots, u_p\}$, we have $z_v = z_v^*$.

Observe that

$$\Gamma(z, z^*) = \{(i,j): i \neq j \text{ and } (z_i, z_j) \neq (z_i^*, z_j^*)\} = S_1 \cup S_2,$$

where

$$S_1 = \{(i,j): z_i \neq z_i^* \text{ and } j \neq i\}$$
$$S_2 = \{(i,j): z_i = z_i^* \text{ and } z_j \neq z_j^*\}.$$

Thus, we have

$$\sum_{\substack{i < j \\ (i,j) \in \Gamma(z,z^*)}} (1-t)\text{Ren}_t\left(P_{ij}^z, P_{ij}^*\right) = \frac{1}{2} \sum_{(i,j) \in \Gamma(z,z^*)} (1-t)\text{Ren}_t\left(P_{ij}^z, P_{ij}^*\right)$$

$$= \frac{1}{2}\left(\underbrace{\sum_{(i,j) \in S_1} (1-t)\text{Ren}_t\left(P_{ij}^z, P_{ij}^*\right)}_{T_1(t)} + \underbrace{\sum_{(i,j) \in S_2} (1-t)\text{Ren}_t\left(P_{ij}^z, P_{ij}^*\right)}_{T_2(t)}\right).$$

Notice further that

$$T_1(t) = \sum_{i \in \{u_1, \cdots, u_m\}} \sum_{j \neq i} (1-t)\text{Ren}_t\left(P_{ij}^z, P_{ij}^*\right) \tag{C.4}$$

and

$$T_2(t) = \sum_{i \notin \{u_1, \cdots, u_m\}} \sum_{j \in \{u_1, \cdots, u_m\}} (1-t)\text{Ren}_t\left(P_{ij}^z, P_{ij}^*\right)$$

$$= \sum_{i \in \{u_1, \cdots, u_m\}} \sum_{j \notin \{u_1, \cdots, u_m\}} (1-t)\text{Ren}_t\left(P_{ij}^z, P_{ij}^*\right)$$

$$= T_1(t) - \sum_{i \in \{u_1, \cdots, u_m\}} \sum_{\substack{j \in \{u_1, \cdots, u_m\} \\ j \neq i}} (1-t)\text{Ren}_t\left(P_{ij}^z, P_{ij}^*\right).$$

Combined to the Chernoff bounds (C.2), this leads

$$\mathbb{P}\left(L(z) \geq L(z^*)\right) \leq e^{-T_1(t)+T_3(t)}, \tag{C.5}$$

where $T_3(t)$ is given by

$$T_3(t) = \frac{1}{2} \sum_{\substack{i \in \{u_1, \cdots, u_m\}}} \sum_{\substack{j \in \{u_1, \cdots, u_m\} \\ j \neq i}} (1-t)\mathrm{Ren}_t\left(P_{ij}^z, P_{ij}^*\right). \tag{C.6}$$

Let us lower-bound $T_1$. For $p \in [m]$, denote $t_p = \arg\max_{t \in (0,1)} (1-t) \sum_{j \neq u} \mathrm{Ren}_t\left(P_{u_p j}^z, P_{u_p j}^*\right)$. Note that $t_p$ is bounded away from one, as when $t = 1$, the objective function inside the argmax equals 0. We also recall that, for any $\alpha, \beta \in (0,1)$ with $\alpha \leq \beta$, and any probability distributions $f$ and $g$, we have (Van Erven and Harremoës, 2014, Theorem 16)

$$\frac{\alpha}{\beta}\frac{1-\beta}{1-\alpha}\mathrm{Ren}_\beta(f,g) \leq \mathrm{Ren}_\alpha(f,g) \leq \mathrm{Ren}_\beta(f,g).$$

Denote $t^* = \min\{t_1, \cdots, t_m\}$. Without loss of generality, suppose that $t^* = t_1$. Using the previous inequality with $\alpha = t_p$ and $\beta = t_1$, we have,

$$\sum_{j \in [n] \setminus \{u_p\}} \mathrm{Ren}_{t_1}\left(P_{u_p j}^z, P_{u_p j}^*\right) \geq \sum_{j \in [n] \setminus \{u_p\}} \mathrm{Ren}_{t_p}\left(P_{u_p j}^z, P_{u_p j}^*\right),$$

for any $p \in [m]$. Thus,

$$(1-t_1) \sum_{j \in [n] \setminus \{u_p\}} \mathrm{Ren}_{t_1}\left(P_{u_p j}^z, P_{u_p j}^*\right) \geq \frac{1-t_1}{1-t_p}(1-t_p) \sum_{j \in [n] \setminus \{u_p\}} \mathrm{Ren}_{t_p}\left(P_{u_p j}^z, P_{u_p j}^*\right)$$

$$= \frac{1-t_1}{1-t_p}\mathrm{Chernoff}(u_p),$$

by definition of $t_p$. Because all the $t_p$ are bounded away from 1 and $t_1 = \min\{t_1, \cdots, t_m\}$, we have $\frac{1-t_1}{1-t_p} \geq C$ for some constant $C \geq 1$. Recalling the definition of $T_1$ in (C.4), we obtain

$$T_1(t_1) \geq \sum_{p=1}^{m} \frac{1-t_1}{1-t_p}\mathrm{Chernoff}(u_p)$$

$$= \mathrm{Chernoff}(u_1) + \sum_{p=2}^{m} \frac{1-t_1}{1-t_p}\mathrm{Chernoff}(u_p)$$

$$\geq \mathrm{Chernoff}(u_1) + C \sum_{p=2}^{m} \mathrm{Chernoff}(u_p). \tag{C.7}$$

We now upper-bound $T_3(t_1)$, defined in (C.6). By Assumption 1, all the Rényi divergences are of the same order. Thus, there exists a quantity $C_m'$ such that $C_n' = 1$ and

$$T_3(t_1) \leq \frac{1}{2} \sum_{i \in \{u_1, \cdots, u_m\}} C_m' \frac{m-1}{n} \sum_{j \neq i} (1-t_1)\mathrm{Ren}_{t_1}\left(P_{ij}^z, P_{ij}^*\right)$$

$$\leq C_m' \frac{m}{2n} \sum_{p \in \{1, \cdots, m\}} \mathrm{Chernoff}(u_p, z^*). \tag{C.8}$$

In the rest of the proof, we denote $\delta_m = C_m' \frac{m}{2n}$.

By combining (C.7) and (C.8) with the Chernoff bound (C.5), we have

$$\mathbb{P}\left(L(z) \geq L(z^*)\right) \leq e^{-\mathrm{Chernoff}(u_1)} e^{-(C-\delta_m/2) \sum_{p=2}^{m} \mathrm{Chernoff}(u_p)}.$$

We recall that $z \in \mathscr{Z}_m$ if and only if there exists a set $\{u_1, \cdots, u_m\}$ of $m$ distinct vertices such that

$$z_u = z_u^* \iff u \notin \{u_1, \cdots, u_m\}.$$

Moreover, for any such set $\{u_1, \cdots, u_m\}$, there exists $(k-1)^m$ ways to construct a $z \in \mathscr{Z}_m$. Hence,

$$\sum_{z \in \mathscr{Z}_m} \mathbb{P}\left(L(z) \geq L(z^*)\right) \leq (k-1)^m \sum_{\{u_1, \cdots, u_m\}} e^{-\text{Chernoff}(u_1)} e^{-(C-\delta_m)\sum_{p=2}^m \text{Chernoff}(u_p)}$$

$$= (k-1)^m \sum_{u_1=1}^n e^{-\text{Chernoff}(u_1)} \sum_{\{u_2, \cdots, u_m\}} e^{-(C-\delta_m)\sum_{p=2}^m \text{Chernoff}(u_p)}.$$

In the previous inequality, the second summation is over all set $\{u_2, \cdots, u_m\}$ of $m-1$ elements belonging to $[n] \backslash \{u_1\}$. There are

$$\binom{n-1}{m-1} \leq \left(\frac{e(n-1)}{m-1}\right)^{m-1}$$

ways of choosing such set. We finally obtain

$$\sum_{z \in \mathscr{Z}_m} \mathbb{P}\left(L(z) \geq L(z^*)\right) \leq \sum_{u_1=1}^n e^{-\text{Chernoff}(u_1)} \left(\frac{e(k-1)(n-1)}{m-1} e^{-(C-\delta_m)\min_{i\in[n]}\text{Chernoff}(i)}\right)^{m-1}$$

$$\leq \sum_{u_1=1}^n e^{-\text{Chernoff}(u_1)} \left(\frac{ekn}{m-1} e^{-(C-\delta_m)\min_{i\in[n]}\text{Chernoff}(i)}\right)^{m-1}.$$

**Ending the proof.** Going back to (C.1), we have

$$\mathbb{E}\left[\text{loss}(z^*, \hat{z})\right] \leq \frac{1}{n}\left(m_0 + \sum_{u_1=1}^n e^{-\text{Chernoff}(u_1)} \sum_{m=m_0+1}^{n(1-1/k)} m Q_m^{m-1}\right), \tag{C.9}$$

where $Q_m = \frac{ekn}{m-1} e^{-(C-\delta_m)\min_{i\in[n]}\text{Chernoff}(i)}$. Denote also $R = \sum_{u_1=1}^n e^{-\text{Chernoff}(u_1)}$ and $B = 2enke^{-C\min_i \text{Chernoff}(i,z^*)}$, and recall $C \geq 1$. We also introduce

$$m_1 = \lfloor 2enke^{-(C-\delta_{n(1-1/k)})\min_i \text{Chernoff}(i,z^*)}\rfloor.$$

By assumption, we have $\delta_{n(1-1/k)} < 1 - \epsilon$ and thus $m_1 = o(n)$. Observe that

$$Q_m \leq \frac{1}{2} \quad \forall m \in \{m_1+1, \cdots, n(1-1/k)\}$$

and thus

$$\sum_{m=m_1+1}^{n(1-1/k)} m Q_m^{m-1} \leq \sum_{m=m_1+1}^\infty m\left(\frac{1}{2}\right)^{m-1} \leq 4\frac{m_1+1}{2^{m_1}} \leq 4 \tag{C.10}$$

by using properties on geometric sums (see Lemma 5).

We still need to upper-bound $\sum_{m=m_0+1}^{m_1} m Q_m^{m-1}$. Let $\tilde{m}_0 = 2ekRe^{C\delta_{m_1}\min_i \text{Chernoff}(i,z^*)}$. Observe that, for any $\tilde{m}_0 \leq m \leq m_1$, we have $Q_m \leq 1/2$. Then, we are left with two cases.

(a) If $\tilde{m}_0 \leq 1$, then chose $m_0 = 0$. Then, we simply have

$$\sum_{m=m_0}^{m_1} m Q_m^{m-1} = \sum_{m=1}^n m Q_m^{m-1} \leq \sum_{m=1}^\infty m\left(\frac{1}{2}\right)^{m-1} \leq 4,$$

by using Lemma 5 as above. By combining (C.9) and (C.10), we have

$$\mathbb{E}\left[\text{loss}(z^*, \hat{z})\right] \leq 8\frac{R}{n}.$$

(b) Otherwise, chose $m_0 = \lceil \tilde{m}_0 \rceil$. Then, we upper-bound $\sum_{m=m_0}^{m_1} m Q_m^{m-1}$ by 4 as above, and we obtain from (C.9) that

$$\mathbb{E}\left[\text{loss}(z^*, \hat{z})\right] \leq \frac{1}{n}(m_0 + 8R)$$

Moreover, $m_0 = \lceil \tilde{m}_0 \rceil \leq 2\tilde{m}_0$. This gives

$$\mathbb{E}\left[\text{loss}(z^*, \hat{z})\right] \leq \frac{1}{n}(2\tilde{m}_0 + 8R) \leq \frac{R}{n}\left(2eke^{C\delta_{m_1}\min_i \text{Chernoff}(i,z^*)} + 8\right).$$

Observe that this last upper-bound is also an upper-bound for $\mathbb{E}\left[\text{loss}(z^*, \hat{z})\right]$ in the case (a). To finish the proof, we recall that $m_1 = o(n)$ and thus $\delta_{m_1} = o(1)$ by definition of $\delta_m$. $\quad\square$

**Additional Lemma** This lemma and its proof are taken from (Avrachenkov et al., 2022, Lemma A.8), and reproduced here for the sake of completeness.

**Lemma 5.** *For any integer $M \geq 1$ and any number $0 \leq s < 1$,*

$$Ms^M \leq \sum_{m=M}^{\infty} ms^m \leq (1-s)^{-2}Ms^M.$$

*Proof.* Denote $S = \sum_{m=M}^{\infty} ms^m$. By differentiating $\sum_{m=M}^{\infty} s^m = (1-s)^{-1}s^M$ with respect to $s$, we find that

$$s^{-1}S = \sum_{m=M}^{\infty} ms^{m-1} = (1-s)^{-2}s^M + (1-s)^{-1}Ms^{M-1},$$

from which we see that

$$S = s(1-s)^{-2}\left(s^M + (1-s)Ms^{M-1}\right) = \frac{Ms^M}{(1-s)^2}\left(1 - s(1-1/M)\right)$$

The upper bound now follows from $1 - s(1-1/M) \leq 1$. The lower bound is immediate, corresponding to the first term of the nonnegative series. $\quad\square$

# D    Proof of Proposition 3 and Examples 1 and 2

## D.1    Chernoff divergence for Homogeneous PABM

We start with the following lemma.

**Lemma 6.** *Consider a PABM with homogeneous interactions, and $k$ equal-size communities. Suppose the coefficients $\lambda_1^{\text{in}}, \cdots, \lambda_n^{\text{in}}$ (resp., $\lambda_1^{\text{out}}, \cdots, \lambda_n^{\text{out}}$) are sampled iid from a distribution $\mathcal{D}_{\text{in}}$ (resp., $\mathcal{D}_{\text{out}}$), where $\mathcal{D}_{\text{in}}$ and $\mathcal{D}_{\text{out}}$ are two distributions supported on $\mathbb{R}_+$ and with mean 1. Let $i \in [n]$. We have*

$$\text{Chernoff}(i, z^*) = (1 + o(1))\frac{n\rho_n}{k}\mathbb{E}\left[\left(\sqrt{p_0\lambda_i^{\text{in}}Y} - \sqrt{q_0\lambda_i^{\text{out}}Y'}\right)^2\right],$$

*where $Y$ and $Y'$ are two independent random variables sampled from $\mathcal{D}_{\text{in}}$ and $\mathcal{D}_{\text{out}}$, respectively.*

*Proof.* We apply the law of large number to the quantity $\delta$ defined in the equation above Proposition 3. $\quad\square$

**Lemma 7.** *Consider the same setting and notations as in Lemma 6. We also suppose that the distributions $\mathcal{D}_{\text{in}}$ and $\mathcal{D}_{\text{out}}$ have pdf $f_{\mathcal{D}_{\text{in}}}$ and $f_{\mathcal{D}_{\text{out}}}$ with respect to the Lebesgue measure. Denote $\gamma_{\text{in}} = \mathbb{E}[\sqrt{Y}]$ and $\gamma_{\text{out}} = \mathbb{E}[\sqrt{Y'}]$, where $Y \sim \mathcal{D}_{\text{in}}$ and $Y' \sim \mathcal{D}_{\text{out}}$. Finally, suppose that $p_0 > 0$ and let $\xi = q_0/p_0$. We have*

$$\frac{1}{n}\sum_{i=1}^{n}\exp\left(-\frac{n\rho_n}{k}p_0\left(1 + \lambda_i^{\text{in}} - 2\gamma\sqrt{\lambda_i^{\text{in}}}\right)\right) = (1 + o(1))J_n,$$

*where*

$$J_n = \int\int \exp\left(-\frac{n\rho_n}{k}p_0\left(x + \xi y - 2\gamma_{\text{in}}\gamma_{\text{out}}\sqrt{\xi}\sqrt{xy}\right)\right) f_{\mathcal{D}_{\text{in}}}(x) f_{\mathcal{D}_{\text{out}}}(y) \mathrm{d}x \mathrm{d}y.$$

*Proof.* From Lemma 6, we have

$$\text{Chernoff}(i, z^*) = (1 + o(1))\frac{n\rho_n}{k}p_0 \mathbb{E}\left[\left(\sqrt{\lambda_i^{\text{in}}Y} - \sqrt{\xi\lambda_i^{\text{out}}Y'}\right)^2\right]$$

$$= (1 + o(1))\frac{n\rho_n}{k}p_0\left(\lambda_i^{\text{in}} + \xi\lambda_i^{\text{out}} - 2\gamma_{\text{in}}\gamma_{\text{out}}\sqrt{\xi}\sqrt{\lambda_i^{\text{in}}\lambda_i^{\text{out}}}\right).$$

As $\lambda_i^{\text{in}} \sim \mathcal{D}_{\text{in}}$ and $\lambda_i^{\text{out}} \sim \mathcal{D}_{\text{out}}$, computing $\frac{1}{n}\sum_i e^{-\text{Chernoff}(i,z^*)}$ resumes to compute

$$\lim_{i\to\infty} \frac{1}{n}\sum_{i=1}^n \exp\left(-\frac{n\rho_n}{k}p_0\left(x + \xi y - 2\gamma_{\text{in}}\gamma_{\text{out}}\sqrt{\xi}\sqrt{xy}\right)\right).$$

In particular, $\exp\left(-\frac{n\rho_n}{k}p_0\left(x + \xi y - 2\gamma_{\text{in}}\gamma_{\text{out}}\sqrt{\xi}\sqrt{xy}\right)\right)$ is bounded by $[0, 1]$, hence its variance is also upper bounded by 1. Let $J_n$ be the expectation of this quantity over $\mathcal{D}_{\text{in}}, \mathcal{D}_{\text{out}}$. Centering the variable, we bound the total variance

$$\sum_{i=1}^n n^{-2}\,\text{Var}\left(\exp\left(-\frac{n\rho_n}{k}p_0\left(x + \xi y - 2\gamma_{\text{in}}\gamma_{\text{out}}\sqrt{\xi}\sqrt{xy}\right)\right) - J_n\right) < \sum_i n^{-2} < \infty.$$

Kolmogorov's variance criterion for averages (Kallenberg, 2021, Lemma 5.22) implies

$$\frac{1}{n}\sum_{i=1}^n\left(\exp\left(-\frac{n\rho_n}{k}p_0\left(x + \xi y - 2\gamma_{\text{in}}\gamma_{\text{out}}\sqrt{\xi}\sqrt{xy}\right)\right) - J_n\right) \xrightarrow{\text{a.s.}} 0.$$

Therefore the limit converges to its expectation almost surely,

$$\lim_{i\to\infty} \frac{1}{n}\sum_{i=1}^n \exp\left(-\frac{n\rho_n}{k}p_0\left(x + \xi y - 2\gamma_{\text{in}}\gamma_{\text{out}}\sqrt{\xi}\sqrt{xy}\right)\right) = J_n,$$

where

$$J_n = \int\int \exp\left(-\frac{n\rho_n}{k}p_0\left(x + \xi y - 2\gamma_{\text{in}}\gamma_{\text{out}}\sqrt{\xi}\sqrt{xy}\right)\right) f_{\mathcal{D}_{\text{in}}}(x) f_{\mathcal{D}_{\text{out}}}(y) \mathrm{d}x \mathrm{d}y.$$

$\square$

### D.2   Proof of Proposition 3

To prove Proposition 3, we apply Lemma 7 in the particular case where $\mathcal{D}_{\text{in}}$ is the uniform distribution $\text{Uni}(1 - c, 1 + c)$ and $\mathcal{D}_{\text{out}}$ is the Dirac distribution at 1. Hence, the integral $J_n$ given in Lemma 7 becomes

$$J_n = \int \exp\left(-\frac{n\rho_n}{k}p_0\left(x + \xi - 2\gamma_{\text{in}}\sqrt{\xi}\sqrt{x}\right)\right) f_{\mathcal{D}_{\text{in}}}(x) \mathrm{d}x,$$

where $f_{\mathcal{D}_{\text{in}}}(x) = \frac{1}{2c}\mathbb{1}(x \in (1 - c, 1 + c))$, and the lower and upper limits of the integral are $1 - c$ and $1 + c$, respectively. For simplicity we write $y = \sqrt{Mx}$ where $M = n\rho_n p_0/k$. We perform the following change of variable: $\sqrt{x} = \frac{y}{\sqrt{M}}$, $\mathrm{d}y = \frac{1}{2}\sqrt{M}x^{-1/2}\mathrm{d}x$ and $\mathrm{d}x = \frac{2\sqrt{x}}{\sqrt{M}}\mathrm{d}y = \frac{2y}{M}\mathrm{d}y$. The lower and upper integration limits become $y_- = \sqrt{M}\sqrt{1 - c}$ and $y_+ = \sqrt{M}\sqrt{1 + c}$. Changing

variables and completing the square gets us

$$
\begin{aligned}
J_n &= \frac{1}{2c} \int_{y_-}^{y_+} \exp\left(-y^2 - M\xi + 2\gamma\sqrt{M}y\sqrt{\xi}\right) \frac{2y}{M}\mathrm{d}y \\
&= \frac{1}{2c} \int_{y_-}^{y_+} \exp\left(-(y - \gamma\sqrt{\xi}\sqrt{M})^2 + M\xi(\gamma_{\mathrm{in}}^2 - 1)\right) \frac{2y}{M}\mathrm{d}y \\
&= \frac{\exp(M\xi(\gamma_{\mathrm{in}}^2 - 1))}{cM} \int_{y_-}^{y_+} \exp\left(-(y - \gamma_{\mathrm{in}}\sqrt{\xi}\sqrt{M})^2\right) y\mathrm{d}y.
\end{aligned}
$$

Again, substitute $u = y - \gamma_{\mathrm{in}}\sqrt{\xi}\sqrt{M}$ to get

$$
\begin{aligned}
J_n &= \frac{\exp(M\xi(\gamma_{\mathrm{in}}^2 - 1))}{cM} \int_{u_-}^{u_+} \exp\left(-u^2\right) \left(u + \gamma_{\mathrm{in}}\sqrt{\xi}\sqrt{M}\right) \mathrm{d}u \\
&= \frac{\exp(M\xi(\gamma_{\mathrm{in}}^2 - 1))}{cM} \left( \int_{u_-}^{u_+} \exp\left(-u^2\right) u\mathrm{d}u + \gamma_{\mathrm{in}}\sqrt{\xi}\sqrt{M} \int_{u_-}^{u_+} \exp\left(-u^2\right) \mathrm{d}u \right),
\end{aligned}
$$

where $u_- = \sqrt{M}(\sqrt{1-c} - \gamma_{\mathrm{in}}\sqrt{\xi})$ and $u_+ = \sqrt{M}(\sqrt{1+c} - \gamma_{\mathrm{in}}\sqrt{\xi})$. The first integral can by solved by-parts, and the later we recognize as the Gauss error function. Hence,

$$
J_n = \frac{\exp(M\xi(\gamma_{\mathrm{in}}^2 - 1))}{cM} \left( \frac{1}{2}(\exp(-u_+^2) - \exp(-u_-^2)) + \frac{1}{2}\gamma_{\mathrm{in}}\sqrt{\xi M}\sqrt{\pi}(\mathrm{erf}(u_+) - \mathrm{erf}(u_-)) \right),
$$

where $\mathrm{erf}(t) = 2/\sqrt{\pi}\int_0^t e^{-t^2}\mathrm{d}t$. Moreover, the quantity $\gamma_{\mathrm{in}} = \mathbb{E}_{Y\sim\mathcal{D}_{\mathrm{in}}}[\sqrt{Y}]$ can be computed explicitly. We obtain $\gamma_{\mathrm{in}} = \frac{1}{2c}\int_{1-c}^{1+c}\sqrt{x}\mathrm{d}x = \frac{1}{3c}\left((1+c)^{\frac{3}{2}} - (1-c)^{\frac{3}{2}}\right)$. We denote this last quantity by $\gamma_c$, to emphasize that it depends only on $c$. $\qquad\square$

## D.3 Discussion Relative to Examples 1 and 2

This involved expression of $J_n$ computed in Proposition 3 is well-behaved and practically interesting for particular values of $\xi$ and $c$. As such, a few remarks are in order.

First (resuming Example 1), when $\xi = 1$, we have $u_- = \sqrt{1-c} - \gamma < 0$ and $u_+ = \sqrt{1+c} - \gamma > 0$ for all $c \in (0,1]$. Moreover, $M \to \infty$ as $n \to \infty$, and $u_\mp \to \mp\infty$. So $J$ simplifies to the much simpler expression

$$
J_n = \frac{\gamma_c}{c}\sqrt{\frac{k\pi}{n\rho_n p_0}} \exp\left(-\frac{n\rho_n}{k}p_0(1 - \gamma_c^2)\right),
$$

which only depends on $c$ (recall $\gamma_c = \frac{1}{3c}\left((1+c)^{\frac{3}{2}} - (1-c)^{\frac{3}{2}}\right)$) and is monotonically decreasing over $c \in (0,1]$. This agrees with the following intuitive fact: as $c$ increases, the higher variance in the popularity heterogeneity aids recovery.

Another interesting case is when we fix $c$ as in Example 2. As $\xi$ increases from $0$ to $1$, $J$ first monotonically increase, then monotonically decrease and approaches $0$ as $\xi \to 1$. In particular, $\xi = 0$ corresponds to disconnected communities, hence clustering is trivial. As $\xi$ increases, the additional inter-cluster edges act as noise to our classification task. On the other hand, a very large $\xi$ allows us to better learn from the popularity patterns as $q_0$ gets closer and closer to $p_0$, and leverage from the variance introduced by $c$. Especially, $u_\mp \to \mp\infty$ when $\xi > \xi_0$ for some constant $c_0 \in (0,1)$. Hence in this regime, classification is easy, as $J_n = \frac{\gamma_c\sqrt{\xi\pi}\exp(-M\xi(1-\gamma_c^2))}{c\sqrt{M}} \to 0$ as $M \to \infty$ and $\gamma_c^2 - 1 < 0$. This phenomena illustrates an interesting duality of the role of inter-cluster edges—they act as noise below a threshold $\xi_0$, yet serves to emphasize the popularity variance introduced by $c$ above the same threshold.

To better illustrate this two phenomenon, we plot in Figure 3 the error rates obtained for homogeneous PABM where $\lambda_i^{\mathrm{out}} = 1$ and the $\lambda^{\mathrm{in}}$ are sampled from $\mathrm{Uni}(1-c, 1+c)$. This illustrate the phenomenon

highlighted by the Examples 1 and 2: (i) the error rate do not vanish when the edge-density signal disappear and (ii) the error rate is not monotonously decreasing with the edge-density signal.

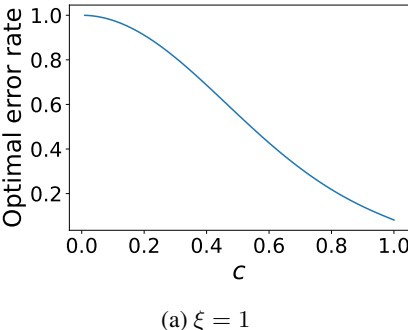

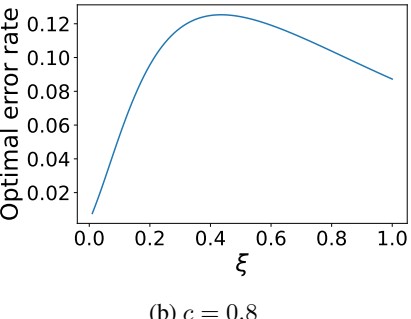

(a) $\xi = 1$          (b) $c = 0.8$

Figure 3: Optimal error rate on PABM with homogeneous interactions. The matrix $P$ is given in Equation (3.1), and we let $n = 900$ vertices, $k = 3$ clusters of same size, average edge density $\rho = 0.05$, and interaction probabilities $p = \rho$ and $q = \xi p$. In both figures, the quantities $\lambda_i^{\text{in}}$ are iid sampled from $\mathcal{D}_{\text{in}} = \text{Uni}(1 - c, 1 + c)$ and the $\lambda_i^{\text{out}}$ are all equal to one. In Figure 3a, we let $\xi = 1$ an vary $c$, while in Figure 3b we let $c = 0.8$ and we vary $\xi$. The optimal error rates are computed using the formula obtained in Proposition 3.

To show that these phenomena are not artifact of setting the $\lambda^{\text{out}}$ all equal to 1 and sampling the $\lambda^{\text{in}}$ from a particular distribution, we also provide in Figure 4 plot of the optimal error rate (as given by the formula derived in Proposition 3) when the coefficients $\lambda^{\text{in}}$ and $\lambda^{\text{out}}$ are sampled from different distributions.

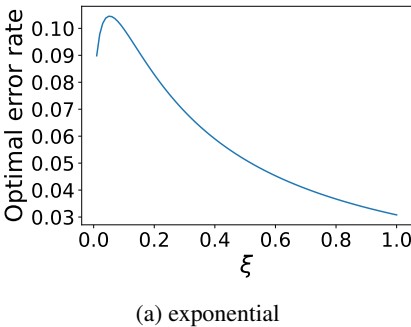

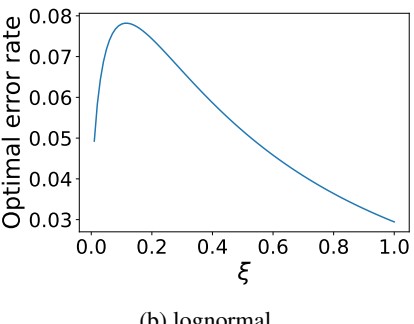

(a) exponential          (b) lognormal

Figure 4: Numerical values obtained for the optimal error rate $\frac{1}{n} \sum_i \exp(-\text{Chernoff}(i, z^*))$ on PABM with homogeneous interactions. The matrix $P$ is given in Equation (3.1), and we let $n = 900$ vertices, $k = 3$ clusters of same size, average edge density $\rho = 0.05$, and interaction probabilities $p = \rho$ and $q = \xi p$. In both figures, the quantities $\lambda_i^{\text{in}}$ and $\lambda^{\text{out}}$ are iid sampled from a distribution $\mathcal{D}$. Figure 4a: $\mathcal{D}$ is the exponential distribution with mean 1. Figure 4b: $\mathcal{D}$ is the log-normal distribution with parameters $(\mu, \sigma) = (-1/2, 1)$ (chosen so that the mean of the distribution is 1).

# E  Description of the Algorithms

## E.1  Variants of Spectral Clustering with $k$ Eigenvectors

Algorithms 1, 2, and 3 provide the *sklearn*, *sbm*, and *dcbm* variants of spectral clustering, respectively.

## E.2  Variants of Spectral Clustering with $k^2$ Eigenvectors

In this section, we describe two algorithms proposed in the litterature for clustering PABM.

**Algorithm 1:** Spectral Clustering: scikit-learn

**Input:** Adjacency matrix $A \in \mathbb{R}_+^{n \times n}$, number of clusters $k$
**Output:** Predicted community memberships $\hat{z} \in [k]^n$

1   Let $D = \mathrm{diag}(D 1_n)$ be the degree matrix
2   Compute the normalized Laplacian $\mathcal{L} = I_n - D^{-1/2} A D^{-1/2}$
3   Compute the $k$ eigenvectors of $\mathcal{L}$ associated to its $k$ smallest eigenvalues. Construct $V \in \mathbb{R}^{n \times k}$ using these eigenvectors as its columns.
4   Let $\hat{z} \in [k]^n$ be the output of Lloyd's algorithm (to solve the $k$-means problem) on the cloud of $k$-dimensional points $V_{i \cdot})_{i \in [n]}$.

---

**Algorithm 2:** Spectral Clustering: standard block model variant

**Input:** Adjacency matrix $A \in \mathbb{R}_+^{n \times n}$, number of clusters $k$
**Output:** Predicted community memberships $\hat{z} \in [k]^n$

1   Compute the $k$ eigenvectors $v_1, \cdots, v_k$ of $A$ associated to its $k$ largest eigenvalues (in absolute value) $|\sigma_1| \geq \cdots \geq |\sigma_k|$. Let $V = (v_1, \cdots, v_k) \in \mathbb{R}^{n \times k}$ and $\Sigma = \mathrm{diag}(\sigma_1, \cdots, \sigma_k) \in \mathbb{R}^{k \times k}$.
2   Let $\hat{z} \in [k]^n$ be the output of Lloyd's algorithm (to solve the $k$-means problem) on the cloud of $k$-dimensional points $((V\Sigma)_{i \cdot})_{i \in [n]}$.

---

**Algorithm 3:** Spectral Clustering: degree-corrected block model variant

**Input:** Adjacency matrix $A \in \mathbb{R}_+^{n \times n}$, number of clusters $k$
**Output:** Predicted community memberships $\hat{z} \in [k]^n$

1   Compute the $k$ eigenvectors $v_1, \cdots, v_k$ of $A$ associated to its $k$ largest eigenvalues (in absolute value) $|\sigma_1| \geq \cdots \geq |\sigma_k|$. Let $V = (v_1, \cdots, v_k) \in \mathbb{R}^{n \times k}$ and $\Sigma = \mathrm{diag}(\sigma_1, \cdots, \sigma_k) \in \mathbb{R}^{k \times k}$.
2   Let $\hat{P} = V \Sigma V^T$
3   Let $S_0 = \{i \in [n] \colon \|P_{i \cdot}\|_1 = 0\}$. Define $\tilde{P}_{i \cdot} = P_{i \cdot} / \|P_{i \cdot}\|_1$ for $i \in S_0^c$ and $\tilde{P}_{i \cdot} = P_{i \cdot}$ for $i \in S_0$.
4   Let $\hat{z} \in [k]^n$ be the output of Lloyd's algorithm (to solve the $k$-means problem) on the cloud of $n$-dimensional points $(\hat{P}_{i \cdot})_{i \in S_0^c}$ (note that we assign the vertices of $S_0$ arbitrarily).

---

**Orthogonal Spectral Clustering**   Koo et al. (2023) observed that PABM is a special case of the Generalized Random Dot Product Graph (GRDPG) for which the latent position vectors lie in distinct orthogonal subspaces, each subspace corresponding to a community. This leads to Algorithm 4.

---

**Algorithm 4:** Orthogonal Spectral Clustering

**Input:** Adjacency matrix $A \in \mathbb{R}_+^{n \times n}$, number of clusters $k$
**Output:** Predicted clusters $\hat{z} \in [k]^n$

1   Compute the eigenvectors of $A$ associated to its $k(k+1)/2$ most positive eigenvalues and $k(k-1)/2$ most negative eigenvalues. Construct $V \in \mathbb{R}^{n \times k^2}$ using these eigenvectors as its columns.
2   Compute $B = |nVV^T| \in \mathbb{R}^{n \times n}$, applying $|\cdot|$ entry-wise.
3   Let $\hat{z} \in [k]^n$ be the output of spectral clustering (see Algorithm 1) applied on the graph whose adjacency matrix is $B$.

---

**Subspace Spectral Clustering**   Noroozi et al. (2021) proposes another approach to cluster PABM. In particular, they notice that the expected adjacency matrix of a PABM has a rank between $k$ and $k^2$ and is composed of subspaces. In particular, two vertices in the same community belong to the same subspace. This motivates the usage of subspace clustering, as opposed to $k$-means, for clustering the cloud of point obtained via the spectral embedding. For subspace clustering, we use the implementation provided in You et al. (2016) and available at `https://github.com/ChongYou/subspace-clustering`, and we refer to Elhamifar and Vidal (2013) for an introduction on (sparse) subspace clustering. We summarized this in Algorithm 5.

---

**Algorithm 5:** Subspace Clustering on Spectral Embedding

---

**Input:** Adjacency matrix $A \in \mathbb{R}_+^{n \times n}$, number of clusters $k$, embedding dimension $d$ (default: $d = k^2$)

**Output:** Predicted clusters $\hat{z} \in [k]^n$

1 Compute the $d$ eigenvectors $v_1, \cdots, v_d$ of $A$ associated to its $d$ largest eigenvalues (in absolute value) $|\sigma_1| \geq \cdots \geq |\sigma_d|$. Construct $V = (v_1, \cdots, v_d) \in \mathbb{R}^{n \times k}$ and $\Sigma = \mathrm{diag}(\sigma_1, \cdots, \sigma_d)$.

2 Let $\hat{z} \in [k]^n$ be the output of *subspace clustering* on the cloud of $d$-dimensional points $((V\Sigma)_{i\cdot})_{i \in [n]}$.

---

### E.3 Additional Clustering Algorithms

The algorithm from Bhadra et al. (2025) is an iterative community detection method designed for the Popularity-Adjusted Block Model (PABM). It begins by computing an adjacency spectral embedding of the network into a low-dimensional space of dimension $d$ (where typically $d = k^2$). For each tentative community, a subspace is estimated via singular value decomposition of the node embeddings in that cluster. The algorithm then greedily reassigns nodes to the community whose subspace yields the smallest projection error, thereby minimizing the objective function. This process iterates until node assignments stabilize, yielding a community structure tailored to the PABM. Although the original paper does not assign a name to the algorithm, we refer to it as Greedy Subspace Projection Clustering (*gspc*). Algorithm 6 provides the pseudo-code.

---

**Algorithm 6:** Greedy Subspace Projection Clustering (*gspc*)

---

**Input:** Adjacency matrix $A \in \mathbb{R}^{n \times n}$, number of communities $K$, embedding dimension $d$ (default: $d = k^2$), initial cluster labels $z^{(0)} \in [k]^n$

**Output:** Final cluster labels $\hat{z} \in [k]^n$

1 Compute adjacency spectral embedding $X \in \mathbb{R}^{n \times d}$ from $A$;

2 Initialize cluster labels $\hat{z} \leftarrow z^{(0)}$;

3 **repeat**

4     **for** $k \leftarrow 1$ **to** $K$ **do**

5         Extract $X_k \leftarrow \{x_i : \ell_i = k\}$;

6         Compute leading $d$ left singular vectors $U_k$ of $X_k$;

7     **for** $i \leftarrow 1$ **to** $n$ **do**

8         **for** $k \leftarrow 1$ **to** $K$ **do**

9             Compute projection loss $L_{ik} \leftarrow \|x_i - U_k U_k^\top x_i\|^2$;

10         Update $\hat{z}_i \leftarrow \arg\min_k L_{ik}$;

11 **until** *no label changes or maximum iterations reached*;

12 **return** $\hat{z}$;

---

Thresholded Cosine Spectral Clustering (*tcsc*), proposed in Yuan et al. (2025), begins by computing the top $k^2$ eigenvectors of the adjacency matrix to capture structural information. Cosine similarities between eigenvector rows are then calculated and thresholded to suppress noise. Finally, Lloyd's algorithm is applied to the thresholded similarity representation to output the predicted cluster labels. Finally, Yuan et al. (2025) also proposes to refine the cluster labels obtained by *tcsc*. This leads to Refined Thresholded Cosine Spectral Clustering (*r-tcsc*), which improve upon the initial labels from *tcsc* by re-estimating block connection probabilities and then reassigning vertices to clusters according to a profile likelihood criterion. This refinement step reduces misclassifications and yields more accurate community recovery. Pseudo-code for *tcsc* is provided in Algorithm 7, and the reader is refered to (Yuan et al., 2025, Theorem 2) for the refinement step.

### E.4 Rank Analysis in PABM

For simplicity, let us consider a PABM with $k = 3$ blocks, and suppose that vertices are ordered such that the first $n_1$ vertices are in the first cluster, the next $n_2$ vertices are in the second cluster, and the last $n_3 = n - n_1 - n_2$ vertices are in the third cluster. For any vertex $i \in [n]$, we denote by $r_i$ its

---
**Algorithm 7:** Thresholded Cosine Spectral Clustering (*tcsc*)
---
**Input:** Adjacency matrix $A \in \mathbb{R}^{n \times n}$, number of communities $k$
**Output:** Predicted clusters $\hat{z} \in [k]^n$

---
1 Compute the top-$K^2$ eigenvectors of $A$ and form $U \in \mathbb{R}^{n \times K^2}$.
2 For each pair of rows $U_i, U_j$, compute the cosine similarity $S_{ij} = \frac{\langle U_i, U_j \rangle}{\|U_i\| \|U_j\|}$.
3 Apply thresholding: set $S_{ij} = 0$ if $S_{ij} < \tau$, where $\tau$ is a data-driven threshold.
4 Apply Lloyd's algorithm ($k$-means) to the rows of $S$ to obtain the cluster labels $\hat{z}$.

---

rank-indexing of its cluster (that is, $r_i = i$ if $i$ is in cluster 1, $r_i = i - n_1$ if $i$ is in cluster 2, and $r_i = i - n_1 - n_2$ if $i$ is in cluster 3). Denote $\Lambda^{(a,b)}$ the matrix of size $n_a$-by-1 such that $\Lambda^{(a,b)}_{r_i} = \lambda_{ib}$. We also assume that $B_{ab} = p1\{a = b\} + q1\{a \neq b\}$ with $p \neq q$. Then, the matrix $P$ is given by

$$P = \begin{pmatrix} p\Lambda^{(1,1)}(\Lambda^{(1,1)})^T & q\Lambda^{(1,2)}(\Lambda^{(2,1)})^T & q\Lambda^{(1,3)}(\Lambda^{(3,1)})^T \\ q\Lambda^{(2,1)}(\Lambda^{(1,2)})^T & p\Lambda^{(2,2)}(\Lambda^{(2,2)})^T & q\Lambda^{(2,3)}(\Lambda^{(3,2)})^T \\ q\Lambda^{(3,1)}(\Lambda^{(1,3)})^T & q\Lambda^{(3,2)}(\Lambda^{(2,3)})^T & p\Lambda^{(3,3)}(\Lambda^{(3,3)})^T \end{pmatrix}.$$

Thus, the matrix $P$ is composed of $k^2 = 9$ blocks of rank one. Excluding trivial cases, the rank of $P$ can take any value between $k = 3$ and $k^2 = 9$. For example, if all the vectors $\Lambda^{(a,b)}$ are all-1 vectors, then $P$ has rank 1. But, if $\Lambda^{(1,1)}$ contains entries that are not all equal to 1, the rank of $P$ increases to 4. Similarly, if both $\Lambda^{(1,1)}$ and $\Lambda^{(1,2)}$ contain non-constant entries, the rank of $P$ becomes 5, and so on.

# F    Additional Numerical Experiments

## F.1    Performance of *tcsc* and *gspc*

In this section, we compare the accuracy obtained by *tcsc* and *gspc* with the accuracy of *pabm* and *osc* (and of *sklearn* as a baseline). We sample PABM with homogeneous interactions, and take the same parameters as in Section 3.1.

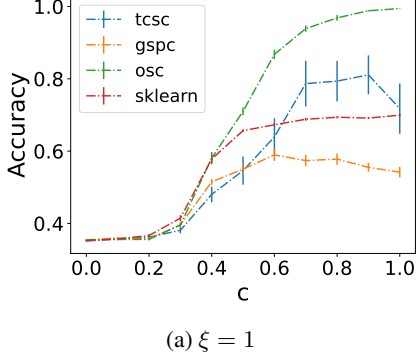
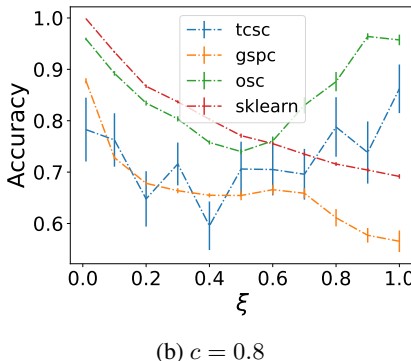

(a) $\xi = 1$                 (b) $c = 0.8$

Figure 5: Performance of graph clustering on homogeneous PABM, where the matrix $P$ is given in Equation (3.1). We sampled graphs with $n = 900$ vertices in $k = 3$ clusters of same size, average edge density $\rho = 0.05$. In both figures, the $\lambda_i^{\text{in}}$ are iid sampled from $\text{Uni}(1 - c, 1 + c)$ and $\lambda_i^{\text{out}} = 1$ for all $i$. Accuracy is averaged over 15 realizations, and error bars show the standard errors.

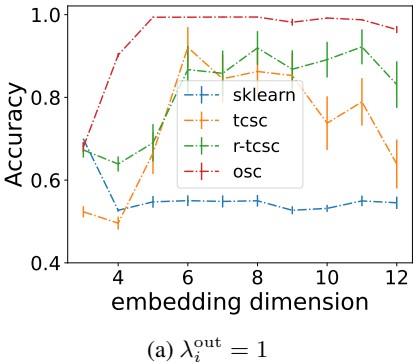
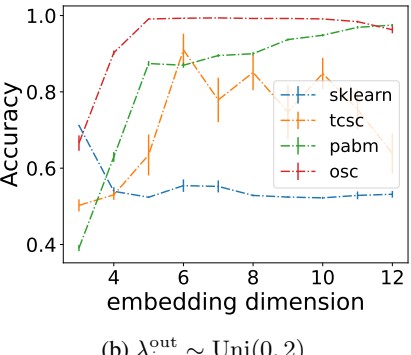

(a) $\lambda_i^{\text{out}} = 1$    (b) $\lambda_i^{\text{out}} \sim \text{Uni}(0, 2)$

Figure 6: Effect of the embedding dimension on the performance of graph clustering on homogeneous PABM, where the matrix $P$ is given in Equation (3.1). We sampled graphs with $n = 900$ vertices in $k = 3$ clusters of same size, average edge density $\rho = 0.05$. In both figures, the $\lambda_i^{\text{in}}$ are iid sampled from $\text{Uni}(0, 2)$. Accuracy is averaged over 15 realizations, and error bars show the standard errors.

## F.2 Numerical Experiments on Heterogeneous PABM

We generate the coefficients $(\lambda_{ia})_{i\in[n],a\in[k]}$ independently from each other and from a distribution with mean 1 and bounded support so that $\sup_{i,a} \lambda_{ia} < 1/\sqrt{\rho}$, and let

$$P_{ij} = \begin{cases} \lambda_{iz_j^*}\lambda_{jz_i^*}\rho & \text{if } z_i^* = z_j^*, \\ \lambda_{iz_j^*}\lambda_{iz_i^*}\xi\rho & \text{otherwise.} \end{cases} \tag{F.1}$$

To generate the $\lambda_{ia}$, we consider the following three distributions: Pareto with exponent $1.5$, log-normal with location $0$ and shape $1$ and exponential with parameter $1$. The support of these distributions is unbounded. To avoid having values too low and too large for the coefficients $\lambda_{ia}$, we sample a random variable $v_{ia}$ following one of these three distributions, and let

$$\lambda_{ia} = \begin{cases} v_{ia} & \text{if } v_{ia} \in [\tau_{\min}, \tau_{\max}], \\ \tau_{\min} & \text{if } v_{ia} < \tau_{\min}, \\ \tau_{\max} & \text{if } v_{ia} > \tau_{\max}. \end{cases}$$

In all experiments, we set $\tau_{\min} = 0.05$ and $\tau_{\max} = 5$. Finally, we normalize the $\lambda_{ia}$ to ensure that $\sum_i \lambda_{ia} = 1$ for all $a \in [k]$. Figure 7 show that *pabm* and *osc* almost always outperform the *sbm* and *dcbm* variants.

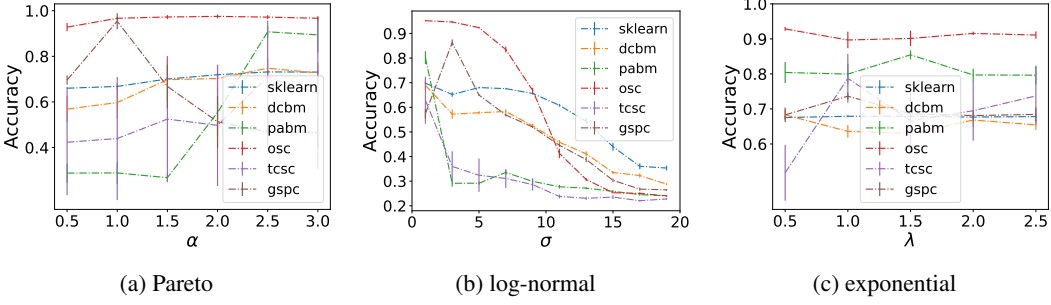

(a) Pareto    (b) log-normal    (c) exponential

Figure 7: Performance of clustering algorithms on heterogeneous PABM, where the matrix $P$ is given in (F.1) with $\rho = 0.05$, and the $\lambda_{ia}$ coefficients are sampled as described in the text. The curve show the average accuracy on 10 realization of PABM with $n = 2000$ vertices in $k = 5$ clusters of same size. Error bars show the standard errors (over 15 realizations).

### F.3 Real Data Sets Description

Table 3 provides some statistics about the graph used. For all graphs, we only considered the largest connected components. Moreover, for *LiveJournal* data set, we extract the two largest clusters. Finally, for MNIST, FashionMNIST and Cifar10, we first embed the images into a low-dimensional space and we consider the $k$-nearest neighbor graph (with $k = 10$) obtained from $n = 10,000$ images. We use the embedding provided in the *graphlearning* package.[4]

| data set | $n$ | $|E|$ | $k$ | $\bar{d}$ | $\sqrt{\overline{d^2} - (\bar{d})^2}$ | Reference |
|---|---|---|---|---|---|---|
| political blog | 1,222 | 16,714 | 2 | 27.3 | 38.4 | Adamic and Glance (2005) |
| LiveJournal-top2 | 2,766 | 24,138 | 2 | 17.5 | 31.8 | Backstrom et al. (2006) |
| citeseer | 2,110 | 3,668 | 6 | 3.5 | 4.0 | Getoor (2005) |
| cora | 2,485 | 5,069 | 7 | 4.1 | 5.4 | Getoor (2005) |
| MNIST | 10,000 | 85,938 | 10 | 17.2 | 5.0 | LeCun et al. (1998) |
| FashionMNIST | 10,000 | 83,486 | 10 | 16.7 | 4.0 | Xiao et al. (2017) |
| CIFAR-10 | 10,000 | 97,044 | 10 | 19.4 | 8.8 | Krizhevsky et al. (2009) |

Table 3: Summary of some statistics of the real data sets considered. The quantities $n$, $|E|$, and $k$ refer to the number of vertices $n$, of edges, and of clusters. The quantities $\bar{d}$ and $\sqrt{\overline{d^2} - (\bar{d})^2}$ refer to the average and standard deviation of the degrees, respectively.

---

[4]`https://pypi.org/project/graphlearning/`.

