# OpenReview forum: "Optimal Graph Clustering without Edge Density Signals"
_NeurIPS.cc/2025/Conference — NeurIPS 2025 poster_

### Official Review · Reviewer_r5oC · 2025-06-29

**Clarity:** 2
**Significance:** 3
**Originality:** 4
**Rating:** 5
**Confidence:** 4

**Summary:**

This paper concerns with the optimal error rate of Popularity Adjusted Block Model (PABM), which is a generalization of Degree-Corrected Block Model (DCBM). In a DCBM, each vertex has its popularity parameter which affects its connection probability to everyone else. In a PABM of k communities, there will be k such parameters for a specific vertex, which affect its connection probability to vertices in difference communities correspondingly. An optimal error rate is identified with matching upper and lower bound. Empirical studies include a few existing algorithms, which verify a point that looking at the top k^2 eigenvectors might be the right algorithm for spectral clustering on PABM with k communities.

**Questions:**

**Q1:** What is the conclusion when Assumption 2 does not hold?

I would naturally expect either the optimal error rate identified by Theorem 1&2 holds even when the divergence is constant order, which would reflect a constant error rate as the prior work on DCBM [Gao et al. ‘18] has their minimax rate holds even to the constant average degree regime (and thus only a constant fraction of recovery is possible), or there is a negative result saying that when Assumption 2 does not hold, the optimal error rate is at least some constant, like in [Zhang and Zhou ‘16]. The Chernoff-Hellinger divergence for SBM also characterizes the information-theoretic error rate when exact recovery is not possible but almost exact recovery is possible (an estimator has o(n) errors with high probability).

**Q2:** What is the assumption on the dependency between matrix $B$ and $p,q$?

I understand that $B$ takes only $p$ and $q$ when the model is homogeneous (in this case, $\lambda_{iz_j}$ takes two possible values as well). Is the matrix $B$ having diagonal values greater than or equal to $p$ and the off-diagonal values less than or equal to $q$ ($\*$), or is it the case that the result holds even without this constraint? My understanding is the rate identified in Theorem 1 & 2 does not depend on any assumptions on $B$, putting assumptions on the dependency between $B$ and $p,q$ would give it a specific formula, like the perfect homogeneity gives the formula between line 88 and line 89. Does the same specific rate hold for the ($\*$) case I described above?

**Q3:** Can the main result be formalized into a minimax rate or an information-theoretic result? How should I understand their relations to the result in this paper?

I was trying to understand this optimal error rate result as something quite independent to those minimax rates and information-theoretical analysis, like the result in [1]. But obviously some minimax rates as well as the Chernoff-Hellinger divergence [Abbe and Sandon '15], which determines the information-theoretic threshold for exact recovery on SBM, can be seemingly read from the main result.

--

[1] Zhang, Anderson Ye. "Fundamental limits of spectral clustering in stochastic block models." IEEE Transactions on Information Theory (2024).

**Ethical Concerns:**

["NO or VERY MINOR ethics concerns only"]

**Final Justification:**

The rebuttal addressed my main concerns so I raised my score.

**Limitations:**

Yes.

**Paper Formatting Concerns:**

No concerns.

**Quality:**

3

**Strengths And Weaknesses:**

**Strengths:**

The authors provide $z^*$ depending matched error bounds, and identify two interesting phenomena in Example 1 and Example 2, which are: *a)* Non-trivial clustering is possible even when $p=q$ and
    *b)* The minimax rate is not monotonically increasing when the number of inter-cluster edges increases.


**Weaknesses:**

1) The result does not exactly match the form of minimax rates in the prior works such as [Zhang and Zhou ‘16] and [Gao et al. ‘18].

2) This work does not provide an efficient algorithm that provably achieves the optimal error rate identified in this work.

3) The opposite side of Assumption 2 is not formally discussed.

4) Some parts of the writing are a bit unclear to me.

    a) In the Introduction, PABM is introduced under the homogeneous setting, which has parameters $p$ and $q$. In Section 2, PABM is defined under the general heterogeneous setting, with matrix B. However, the dependency between these two sets of parameterizations is not formally discussed.

    b) There is no formal definition of those various terms of recovery (e.g. consistent recovery, cluster recovery, and recovery). From my understanding, they all refer to $\lim_{n\rightarrow \infty} P(\hat{z}=z^*)=1$ (Just for clarification, I would refer to it as exact recovery in this review).

---

> ### Author Rebuttal · Authors · 2025-07-30
>
> Dear reviewer r5oC,
>
> We thank you for your time in evaluating our submission and we are grateful for your comments. Please find below responses to the questions raised in your review.
>
> 1. The situation when Assumption 2 fails is indeed delicate and deserves a more thorough discussion.
>     - First, we would like to clarify that the results in [Gao et al., 2018] do not fully cover the constant average degree regime. Specifically, their lower-bound (Theorem 2 of Gao et al 2018) assumes that $p \\| \\theta \\|^2_{\\infty} = o(1)$. When $p=\\Theta(1)$, this condition requires $\\| \\theta \\|^2_{\\infty} = o(1)$ which is incompatible with the normalization condition $\\frac{1}{n_a} \\sum_{i : z_i^{\*} = a} \\theta_i \\in (1-\\delta, 1+\\delta)$ imposed on the degree-correction parameters. Their upper bound (Theorem 1 of Gao et al 2018) is more flexible as it only assumes $\\| \theta \\|_{\infty} = o(n/k)$. But, this result still requires $I \rightarrow \infty$ which, in the case of constant $p,q$, implies that the set $S = \{ i \in [n] \colon \theta_i = O(1) \}$ satisfies $|S| = o(n)$. Recalling that the probability of misclustering a vertex $i$ is at least $\exp(-\theta_i( \sqrt{p}-\sqrt{q})^2)$, the set $S$ is composed of the vertices having a non-vanishing probability of being misclassified by the best estimator.
>
>     - Turning to our results, when Assumption 2 fails such that $\max_i \mathrm{Chernoff}(i,z^{\*}) = O(1)$, then the optimal clustering error cannot vanish. Indeed, using arguments similar to those in [Zhang \& Zhou, 2016], we can establish that the optimal error rate is lower-bounded by a non-zero constant $c > 0$. More generally, if we define, analgously to DCBM, the set $S = \\{ i \in [n] \colon \mathrm{Chernoff}(i,z^{\*}) = O(1) \\}$ of vertices having a non-vanishing error of being misclustered. Then, by refining our lower-bound argument, we obtain a lower bound of the form: $\frac{1}{|S^c|} \sum_{i \in S^c} e^{ - \mathrm{Chernoff}(i,z^{\*}) } + c \frac{|S|}{n}$. This decomposition reflects that a constant fraction of nodes (those in $S$) are intrinsically hard to classify, while the rest exhibit standard exponential error decay. When Assumption 2 holds, $|S| = 0$ and the lower bound matches the result in our Theorem 1. We will include this refined lower bound and discussion in the revised version. Extending the upper bound to this more general setting is plausible, but involves technical subtleties that we are currently investigating.
>
> 2. - We do not make any assumption on the matrix $B$ (beyond being symmetric). As a result, our analysis derives the optimal error rate for a specific instance of PABM (similar to the instance-wise analysis in [Yun and Proutière, 2016] for the edge-labeled SBM) rather than a minimax error rate (as in [Zhang \& Zhou, 2016] for SBM and [Gao, Ma, Zhang, Zhou, 2018] for DCBM). Both approaches are valuable: the minimax framework requires defining a parameter space to which $B$ belongs (typically the space (*) mentioned in the review), but offers no guarantees when $B$ lies outside this space, while the instance optimal-rate restricts to a specific but arbitrary matrix $B$.
>     -  Moreover, we wish to emphasize an important point: a rate-optimal algorithm in the minimax setting may not be rate-optimal for specific instances—even when those instances fall within the defined parameter space. For example, Lloyd’s algorithm is minimax-optimal over the class of sub-Gaussian mixture models [Lu \& Zhou, 2016. Statistical and computational guarantees of Lloyd’s algorithm and its variants. arXiv:1612.02099], but it fails to be instance-optimal for Gaussian mixture models with anisotropic covariance structures [Chen \& Zhang, 2024. Achieving optimal clustering in Gaussian mixture models with anisotropic covariance structures. NeurIPS 2024].
>     -   For the parameter space (*), the worst-case rate for SBM and DCBM arises when $B_{aa} = p$ for all $a \in [k]$ and $B_{ab} = q$ for all $a \ne b$, leading to a minimax rate involving the term $(\sqrt{p} - \sqrt{q})^2$. In contrast, for PABM, the situation is more complex because of the additional dependence on the individual parameters $\lambda_{ia}$. As a result, we do not believe a simple closed-form expression for the minimax rate in PABM is attainable. Indeed, as shown in Example 2, reducing the gap between $p$ and $q$ does not necessarily increases the optimal error-rate.
>
> 3. -  To express our result in a minimax setting, one would need to define a parameter space $\Theta$, and analyze the maximum error-rate over all instances of PABM belonging to this space. Because the MLE is (asymptotically) rate-optimal for each instance of PABM, the minimax rate is the maximum error rate over $\Theta$. However, identifying this worst-case rate is, in general, intractable for most meaningful choices of $\Theta$. For this reason, we focused on analyzing specific instances of PABM and did not pursue a full minimax characterization.
>      -   Our results recover known expressions in special cases (for example, the Chernoff-Hellinger divergence in SBM arises from a particular instance of a SBM, and the minimax rate for DCBM is also recovered when all intra-cluster connection probabilities equal $p$ and all inter-cluster probabilities equal $q$). In principle, one could derive information-theoretic results such as thresholds for exact recovery by analyzing when $n \\, \mathrm{loss}(z^*, \hat{z}) \rightarrow 0 $. However, for PABM, this does not lead to a clean, interpretable formula, except in the case of SBM where the threshold is precisely given by the Chernoff-Hellinger divergence.

---

> > ### Comment · Reviewer_r5oC · 2025-08-06
> >
> > Thank you very much for the detailed response. It resolves my main concerns and I will raise my score.

---

### Official Review · Reviewer_fKLt · 2025-06-30

**Clarity:** 4
**Significance:** 3
**Originality:** 3
**Rating:** 5
**Confidence:** 4

**Summary:**

The authors investigate the information-theoretic limits of the almost exact recovery problem under the ​​Popularity-Adjusted Block Model (PABM)​​, a variant of the ​​Degree-Corrected Stochastic Block Model (DCSBM)​​ that accounts for differences in popularity between in-cluster and out-cluster edges. Specifically, they derive an information-theoretic lower bound on the misclassification rate and establish its achievability. Additionally, the authors provide empirical validation of the spectral clustering method applied to this model.

**Questions:**

1) Rank Analysis in Spectral Methods

In the empirical studies, the authors show that the rank $r$ of the proposed model ranges in $[K, K^2]$, whereas for traditional models (e.g., SBM, DCSBM, Gaussian mixtures), this value is exactly K. This necessitates the use of higher-order eigenvectors in spectral methods. The reviewer is curious about how the rank $r$ relates to the underlying parameters of the model—could the authors provide further theoretical or empirical insights into this dependency?

2) Extension of Spectral Method Error Rates

[Zhang, 2024] demonstrates that simple spectral clustering (without MLE) can achieve exponentially small error rates for the naive SBM. The reviewer wonders whether this result can be extended to the proposed PABM.

**Ethical Concerns:**

["NO or VERY MINOR ethics concerns only"]

**Final Justification:**

The authors answered my questions.

**Quality:**

4

**Strengths And Weaknesses:**

Strengths:

The work is solid, and the problem is of significant interest to the research community. The authors establish both the information theoretical limitation and the achievability by leveraging maximum likelihood estimators (MLE). In Theorem $1$, they derive a lower bound on the theoretical limitation using Chernoff divergence, and in Theorem $2$, they demonstrate its achievability up to a constant factor of $(1-\eps)/4$. Additionally, in Section 2.4, the authors elucidate the connections and distinctions between their proposed model and previously established influential models, highlighting the generality of their approach by showing that their results can recover those of earlier work.

Weaknesses:
1) It would be better to highlight the technical challenges  in the proof compare with previous work, especially with [Dreveton et al 2024, Gao et al 2018].

2)  It seems that there is still a constant gap between the converse (lower bound) and achievability (upper bound). Could the authors elaborate more on this gap and compare it with related literatures. i.e , [Dreveton et al 2024, Gao et al 2018, Yun and Proutière 2016].

3) While the achievability is established through the optimal MLE estimator, its practical implementation remains challenging due to severe non-convexity. Previous works typically address this limitation by employing spectral method results as initial estimates, followed by iterative refinement through MLE updates across all nodes.

---

> ### Author Rebuttal · Authors · 2025-07-29
>
> Dear Reviewer fKLt,
>
> We thank you for your time in evaluating our submission and we are grateful for your comments. Please find below responses to the weaknesses and questions raised in your review.
>
> Weaknesses
> 1. We acknowledge that we did not include an overview of the proof techniques (partly due to space constraints) and we will add this in the revised version. While the overall structure of the proofs for Theorems 1 and 2, which establish the optimal error rate, is similar to that used for SBM and DCBM, the PABM setting introduces additional technical complexity that requires a more refined analysis.
>    - For the lower-bound, a first challenge is to address the minimum over all permutations in the definition of the error loss. Hence, rather than directly examining $\inf_{\hat{z}} \mathbb{E}[ n^{-1} \mathrm{loss}(z^{\*}, \hat{z}) ]$, we follow previous work such as [Zhang \& Zhou 2016] and [Gao, Ma, Zhang, Zhou 2018] and focus on a sub-problem $\inf_{ \hat{z} \in \mathcal{Z} } \mathbb{E}[ n^{-1} \mathrm{loss}(z^{\*}, \hat{z}) ]$, where $\mathcal{Z} \subset [k]^n$ is chosen such that $\mathrm{loss}(z^{\*}, \hat{z}) = \mathrm{Ham}( z^{\*}, \hat{z} )$ for all  $z^{\*}, \hat{z} \in \mathcal{Z}$. The idea is that this sub-problem is simple enough to analyze, while still capturing the hardness of the original clustering problem. Next, we use a result from [Dreveton, Gözeten, Grossglauser, Thiran 2024] to show that the Bayes risk $\inf_{\hat{z}_i} \mathbb{P} ( \hat{z}_i \ne z_i^{\*})$ for the misclustering of a single vertex $i$ is asymptotically $e^{-(1+o(1)) \mathrm{ Chernoff}(i,z^{\*})}$. (We refer to our answer of Question 1 of reviewer 2qS9 for an overview of this result.) We acknowledge that this part follows to some extent from existing and established techniques in the literature on optimal error rates in block models and mixture models.
>    - The proof of the upper bound is however more involved, and required a new approach. It begins, similarly to prior work on block models, by upper-bounding $\mathbb{E} [ \mathrm{loss}(z^{\*}, \hat{z}) ]$ (where $\hat{z}$ is the MLE) by $\sum_m \mathbb{P}( \mathrm{Ham}(z^{\*}, \hat{z}) = m)$. Thus, the core difficulty relies in upper-bound the quantities $\mathbb{P}(L(z) > L(z^{\*}))$ for any $z$ such that $\mathrm{Ham}(z^{\*}, z) = m$ (where $L(z)$ denotes the likelihood of $z$ given an observation of $A$). This is more challenging in PABM than in SBM and DCBM. Indeed, unlike in SBM (and to some extent DCBM), the likelihood ratio $\frac{L(z)}{L(z^{\*})}$ cannot be easily simplified. As for SBM and DCBM, we rely on Chernoff bounds to obtain $\mathbb{P}(L(z) > L(z^{\*})) \le \mathbb{E} [ e^{t \log (L(z) /  L(z^{\*}) } ]$ for any $t>0$. But, in SBM and DCBM, one can use $t = 1/2$ and obtain clean exponential bounds whose terms are $\exp(- \theta_u \theta_v (\sqrt{p}-\sqrt{q})^2)$. For PABM, the optimal $t$ to use depends intricately on the misclassified set $\{ u \colon z_u \ne z_u^*\}$, and thus on $z$ itself. To address this, we adopt a more refined approach: we decompose the upper bound into three components $T_1(t)$, $T_2(t)$, and $T_3(t)$, and select a tailored value of $t$ for each labeling $z$. This additional complexity distinguishes our analysis from earlier work and reflects the greater structural richness of PABM compared to SBM and DCBM.
> 2.  A gap between the second-order terms in the lower and upper bounds is unavoidable, as the MLE is not Bayes optimal for the loss function $\mathrm{loss}(z^{\*},\hat{z})$, and is only asymptotically optimal. This phenomenon is also observed in prior works. For instance, in [Zhang \& Zhou, 2016], the error rate for SBM takes the form $\exp( - (1+\eta) nI/(\beta k) )$ where $\eta = o(1)$. However, the sequences $\eta$ appearing in the lower and upper bounds are not necessarily identical. Notably, for any constant $C > 0$, as $I \rightarrow \infty$, we have $C \exp( - (1+\eta) nI/(\beta k) ) = \exp( - (1+\eta') nI/(\beta k) )$ where $\eta' = \eta - \frac{\beta k}{nI} \log C = o(1)$. Thus, a constant multiplicative factor can be absorbed into the $o(1)$ term inside the exponent.  The same reasoning applies [Yun and Proutière, 2016] (see, for example, the remark following Theorem 2: “the number of misclassified items scales at least as $n \exp(-(1 + o(1)) n D(\alpha, p))$”, where the $o(1)$ term is left unspecified), to DCBM [Gao et al., 2018], and to our own results.  In particular, the sequences $\eta$ appearing in Theorems 1 and 2 are not the same, and the multiplicative factor $(1-\epsilon) \min_a \pi_a /4$ can be absorbed into the sequence $\eta$, as this factor is of constant order and the term $\frac{1}{n} \sum_i e^{-\mathrm{Chernoff}(i)}$ goes to zero. We chose to make this term explicit in our bounds to avoid having the sequence $\eta$ depend on the parameter $\epsilon$. In the revised version, we will include a remark following Theorem 1 to clarify that the gap between the lower and upper bounds is purely due to second-order terms.
> 3. We acknowledge that the spectral clustering algorithm used in the numerical section does not come with theoretical guarantees. While a two-step procedure is an appealing direction, it presents a significant challenge in the context of PABM.
>  Specifically, the second step via an iterative refinement requires a consistent estimator of the model parameters (which is derived from the initial clustering). In SBM, this is feasible, because the model is parameterized by $\binom{k}{2}$ edge probabilities (see [Zhang \& Zhou 2016] and [Yun \& Proutière 2016]). However, this becomes impossible in DCBM which involves an additional $N$ individual degree parameters $\theta_1, \cdots, \theta_N$. To address this, [Gao, Ma, Zhang, Zhou 2018] proposed a different strategy: for each vertex $i$, they count the number of neighbors vertex $i$ has in each community, normalize these counts by the corresponding community sizes, and assign $i$ to the community with the highest normalized count. While this method bypasses direct estimation of the model parameters, it critically relies on the assumption that intra-cluster edge density is higher than inter-cluster density. Notably, this refinement breaks down in DCBMs with non-homogeneous interactions, where the connection probabilities $p_{ab}$ across cluster pairs are arbitrary (and we are not aware of any existing work that establishes rate-optimality for a polynomial-time algorithm in such a setting). For the same reason, this refinement also breaks down in PABM with homogeneous interactions where $p=q$. Deriving a rate-optimal polynomial-time algorithm for PABM is a crucial avenue for future research.
>
> Questions:
> 1. For simplicity, let us consider a PABM with $k=3$ blocks of equal-size, and suppose that vertices are ordered such that the first $n_1$ vertices are in the first cluster, the next $n_2$ vertices are in the second cluster, and the last $n_3 = n - n_1 - n_2$ vertices are in the third cluster. For any vertex $i \in [n]$, we denote by $r_i$ its rank-indexing of its cluster (that is, $r_i = i$ if $i$ is in cluster 1, $r_i = i-n_1$ if $i$ is in cluster 2, and $r_i = i-n_1-n_2$ if $i$ is in cluster 3). Denote $\Lambda^{(a,b)}$ the matrix of size $n_a \times 1$ such that $\Lambda_{r_{i}}^{(a,b)}  = \lambda_{i b}$. We also assume that $B_{ab} = p 1(a=b) + q 1(a \ne b)$ with $p \ne q$. Then, the matrix $P$ is given by
>  \begin{align*}
>   P =
>   \begin{pmatrix}
>     p \\Lambda^{(1,1)} (\Lambda^{(1,1)} )^T \&
>     q \Lambda^{(1,2)} (\Lambda^{(2,1)} )^T \&
>     q \Lambda^{(1,3)} (\Lambda^{(3,1)} )^T \\\\
>     q \Lambda^{(2,1)} (\Lambda^{(1,2)} )^T \&
>     p \Lambda^{(2,2)} (\Lambda^{(2,2)} )^T \&
>     q \Lambda^{(2,3)} (\Lambda^{(3,2)} )^T \\\\
>     q \Lambda^{(3,1)} (\Lambda^{(1,3)} )^T \&
>     q \Lambda^{(3,2)} (\Lambda^{(2,3)} )^T \&
>     p \Lambda^{(3,3)} (\Lambda^{(3,3)} )^T
>   \end{pmatrix}.
>  \end{align*}
> Thus, the matrix $P$ is composed of $k^2 = 9$ blocks of rank one. Excluding trivial cases, the rank of $P$ can take any value between $k = 3$ and $k^2 = 9$.  For example, if all the vectors $\Lambda^{(a,b)}$ are all-1 vectors, then $P$ has rank $1$. But, if $\Lambda^{(1,1)}$ contains entries that are not all equal to 1, the rank of $P$ increases to $4$. Similarly, if both $\Lambda^{(1,1)}$ and $\Lambda^{(1,2)}$ contain non-constant entries, the rank of $P$ becomes 5, and so on. We acknowledge that we did not explain in detail how the result $rank(P) \in [k,k^2]$ arises, as we only relied instead on prior works. We plan to provide a more explicit explanation in the revised version.
> 2. This is indeed a natural follow-up question to our work. The version of spectral clustering studied in [Zhang, 2024] involves applying $k$-means to the embedding obtained from the top $k$ eigenvectors, and its optimality has been established only for SBM with homogeneous interactions. However, it is not immediately clear that this version of spectral clustering remains rate-optimal for SBM with heterogeneous interactions. Furthermore, our numerical experiments show that this version of spectral clustering fails to achieve the optimal rate in PABM (and possibly in DCBM as well, where normalization of the embedding (as done in [Gao, Ma, Zhang, Zhou 2018]) is typically required to recover optimal performance).

---

> > ### Comment · Reviewer_fKLt · 2025-08-02
> >
> > Thanks for your informative replies. I have raised my points.

---

### Official Review · Reviewer_2qS9 · 2025-07-01

**Clarity:** 2
**Significance:** 2
**Originality:** 3
**Rating:** 4
**Confidence:** 3

**Summary:**

This paper studies the Popularity-Adjusted Block Model (PABM) that is a variant of the classic Stochastic Block Model (SBM) by introducing popularity for each vertex with other vertices within its own clusters and with vertices in other clusters. The authors establish a lower-bound on the expected clustering error that coincides with the known optimal error-rates for inhomogeneous SBM and Degree-Corrected Block Model (DCBM) when the clusters are of equal size. They also establish an upper bound on the clustering error that is nearly tight with the lower-bound (up to a mild multiplicative factor). Motivated by the high-order interaction of vertices in PABM, they perform numerical experiments to evaluate the effectiveness of spectral clustering methods with various embedding dimensions. The results demonstrate the tradeoff between cluster information and noise in dimension selection.

**Questions:**

-	Q1. It seems that Lemma 2 from the literature (Dreveton et al., 2024) is crucial to the proof of Theorem 1 since it relates the MLE to the Chernoff divergence. What is the intuition behind this lemma? Can you give an intuitive explanation on how to deny all estimators in this proof?
-	Q2. Can you provide high-level descriptions for the proofs of Theorems 1 and 2, and give more technical comparisons to the previous work?
-	Q3. What is the formal definition of the accuracy in the experiments?

**Ethical Concerns:**

["NO or VERY MINOR ethics concerns only"]

**Final Justification:**

I think this paper should be accepted if the authors could clarify the essential contributions of their work.

**Limitations:**

-	The authors have not addressed the limitations, but discussed some open problems in the Conclusion section.
-	The definitions of Renyi divergence and $\otimes$ are missing. The derivation process of the explicit expression of $\Delta(i,a)$ in Line 154 is not given.

**Paper Formatting Concerns:**

No formatting issues.

**Quality:**

3

**Strengths And Weaknesses:**

Strengths:
- S1. It is a solid theoretical paper.
- S2. The optimal lower and upper bounds are strong.

Weaknesses:
- W1. Since some key steps in the proof of the lower bound derive from previous literature, I am a bit confused to the technical contributions of this part. See the questions for more information.
- W2. Because lacking high-level descriptions, the main ideas in the proofs (especially in the lower bound proof) are not clear enough for me.

---

> ### Author Rebuttal · Authors · 2025-07-29
>
> Dear Reviewer 2qS9,
>
> We thank you for your time in evaluating our submission and we are grateful for your comments. Please find below responses to the questions raised in your reviews.
>
> 1. Lemma 2 in [Dreveton, Gözeten, Grossglauser, Thiran 2024] establishes the worst-case error rate for a binary hypothesis testing problem where the observed random variable is drawn from either distribution $f_1$ or $f_2$ (corresponding to hypothesis $H_1$ and $H_2$, respectively). Both $f_1$ and $f_2$ are arbitrary and known probability density functions. By the Neyman–Pearson lemma, the likelihood ratio test (equivalent to the MLE in this context) minimizes the probability of error, thereby ruling out all other estimators for this problem. The error of the MLE is then upper-bounded using Chernoff’s method and lower-bounded using a large deviation argument. In our setting, this hypothesis testing formulation corresponds to Equation (A.1), line 386, where $f_1$ and $f_2$ are product distributions of Bernoulli random variables.
>  We acknowledge that the result at line 390 currently appears without sufficient context, and we will revise the paper to ensure this argument is fully self-contained and clearly motivated.
> 2. We acknowledge that we did not include an overview of the proof techniques (partly due to space constraints) and we will add this in the revised version. While the overall structure of the proofs for Theorems 1 and 2, which establish the optimal error rate, is similar to that used for SBM and DCBM, the PABM setting introduces additional technical complexity that requires a more refined analysis.
>    - For the lower-bound, a first challenge is to address the minimum over all permutations in the definition of the error loss. Hence, rather than directly examining $\inf_{\hat{z}} \mathbb{E}[ n^{-1} \mathrm{loss}(z^{\*}, \hat{z}) ]$, we follow previous work such as [Zhang \& Zhou 2016] and [Gao, Ma, Zhang, Zhou 2018] and focus on a sub-problem $\inf_{ \hat{z} \in \mathcal{Z} } \mathbb{E}[ n^{-1} \mathrm{loss}(z^{\*}, \hat{z}) ]$, where $\mathcal{Z} \subset [k]^n$ is chosen such that $\mathrm{loss}(z^{\*}, \hat{z}) = \mathrm{Ham}( z^{\*}, \hat{z} )$ for all  $z^{\*}, \hat{z} \in \mathcal{Z}$. The idea is that this sub-problem is simple enough to analyze, while still capturing the hardness of the original clustering problem. Next, we use a result from [Dreveton, Gözeten, Grossglauser, Thiran 2024] to show that the Bayes risk $\inf_{\hat{z}_i} \mathbb{P} ( \hat{z}_i \ne z_i^{\*})$ for the misclustering of a single vertex $i$ is asymptotically $e^{-(1+o(1)) \mathrm{ Chernoff}(i,z^{\*})}$. (We refer to our answer of Question 1 above for an overview of this result.) We acknowledge that this part follows to some extent from existing and established techniques in the literature on optimal error rates in block models and mixture models.
>    - The proof of the upper bound is however more involved, and required a new approach. It begins, similarly to prior work on block models, by upper-bounding $\mathbb{E} [ \mathrm{loss}(z^{\*}, \hat{z}) ]$ (where $\hat{z}$ is the MLE) by $\sum_m \mathbb{P}( \mathrm{Ham}(z^{\*}, \hat{z}) = m)$. Thus, the core difficulty relies in upper-bound the quantities $\mathbb{P}(L(z) > L(z^{\*}))$ for any $z$ such that $\mathrm{Ham}(z^{\*}, z) = m$ (where $L(z)$ denotes the likelihood of $z$ given an observation of $A$). This is more challenging in PABM than in SBM and DCBM. Indeed, unlike in SBM (and to some extent DCBM), the likelihood ratio $\frac{L(z)}{L(z^{\*})}$ cannot be easily simplified. As for SBM and DCBM, we rely on Chernoff bounds to obtain $\mathbb{P}(L(z) > L(z^{\*})) \le \mathbb{E} [ e^{t \log (L(z) /  L(z^{\*}) } ]$ for any $t>0$. But, in SBM and DCBM, one can use $t = 1/2$ and obtain clean exponential bounds whose terms are $\exp(- \theta_u \theta_v (\sqrt{p}-\sqrt{q})^2)$. For PABM, the optimal $t$ to use depends intricately on the misclassified set $\{ u \colon z_u \ne z_u^*\}$, and thus on $z$ itself. To address this, we adopt a more refined approach: we decompose the upper bound into three components $T_1(t)$, $T_2(t)$, and $T_3(t)$, and select a tailored value of $t$ for each labeling $z$. This additional complexity distinguishes our analysis from earlier work and reflects the greater structural richness of PABM compared to SBM and DCBM.
> 3. In the theoretical section, we focus on the *loss*, defined in Equation (2.2) as the proportion of misclustered vertices. In contrast, the numerical experiments report the *accuracy*, defined as the proportion of correctly clustered vertices, which is simply one minus the loss. This choice aligns with common clustering evaluation metrics used by practitioners (such as ARI and NMI), which also take the value 1 under perfect recovery. We acknowledge that the definition of accuracy and of the Rényi divergence were missing in the original submission and will include them in the revised version.

---

> > ### Comment · Reviewer_2qS9 · 2025-08-06
> >
> > Thank the authors for the explanations that point out the essential contributions of their work. Hope the authors also clarify these in the revised version. I stick to my positive rating.

---

### Official Review · Reviewer_V4fn · 2025-07-02

**Clarity:** 4
**Significance:** 2
**Originality:** 2
**Rating:** 4
**Confidence:** 4

**Summary:**

The primary goal of this work is to demonstrate that one can exploit the individual vertex popularity parameters to separate the clusters even when the traditional edge density signal may not be present. The authors establish the optimal clustering rate for the PABM, an extension of the stochastic block model that incorporates the vertex popularity parameters. They highlight the tension between the connectivity patterns signal and the individual vertex popularity.

**Questions:**

See the concerns listed under weaknesses. In addition:
- Isn't stating and emphasizing "clustering without edge density signals" misleading? All the information in the model are still in the edge densities. In the example in Section 1, even when $p=q$, the edge densities $P_{ij}$ carry information about the clusters since they are still different for a pair $(i,j)$ when $z_i = z_j$ vs. the opposite case. It is just that the difference is not homogeneous across node-pairs (in contrast to the block model).

**Ethical Concerns:**

["NO or VERY MINOR ethics concerns only"]

**Final Justification:**

The authors pointed to some novel aspects of their work which I initially missed (e.g. the non-monotonic behavior of the error with respect to edge density).

**Limitations:**

Yes

**Quality:**

4

**Strengths And Weaknesses:**

## Strengths

The paper is clearly written with well-organized explanations and includes helpful examples that illustrate key ideas.

The authors show a clear and thoughtful connection between their work and already existing results in the field of optimal clustering. The alignment with known theoretical results for SBM/DCSBM helps reinforce its relevance and correctness.

The paper presents technically solid results, and the theorems are stated clearly with supporting intuition.

## Weaknesses

While the results are correct and technically sound, they lack novelty in the sense that the main conclusions are somewhat expected given prior work, and the proof techniques follow somewhat standard arguments without introducing significant new ideas.

The experimental results are interesting, however, they appear somewhat detached from the theoretical setting (which uses the practically intractable MLE), making it difficult to assess how well the empirical findings validate the paper’s formal guarantees. In addition, a comparison of the spectral clustering methods would be more informative and fair if it more clearly isolated the effect of the embedding dimensionality $k$, allowing for a better understanding of its impact on performance.

---

> ### Author Rebuttal · Authors · 2025-07-30
>
> Dear Reviewer V4fn,
>
> We thank you for your time in evaluating our submission and we are grateful for your comments. Please find below responses to the weaknesses and questions raised in your review.
>
> ### Weaknesses
>
> 1. - The optimal error rate in PABM could, to some extent, have been anticipated by experts in the field. However, while the proof for the lower-bound builds on prior work, establishing the upper-bound involves nontrivial and novel technical components, as detailed in our responses to other questions.
>    -  More importantly, our work goes beyond extending the optimal error rates to a more general model. Indeed, our work reveals two counter-intuitive phenomena that have not been previously observed: (i) the non-monotonic behavior of the clustering error with respect to edge density, and (ii) the possibility to recover the clusters even when $p = q$. These insights, discussed following Proposition 3 and illustrated in Examples 1 and 2, deepen our understanding of the complexity of graph clustering under PABM and were overlooked in earlier work. Finally, the derivations in Proposition 3, even if conducted in a simplified setting, are technically involved and further highlight the subtlety of these phenomena.
>
> 2. - In the numerical section, we illustrate two surprising phenomena discussed in the theoretical section: the non-monotonic behavior of the error with respect to edge density, and the ability to recover clusters even when $p=q$. While it would have been possible to use a greedy algorithm to approximate the MLE, we opted for spectral methods because of their widespread use and established effectiveness for clustering in block models. In particular, our experiments demonstrate that the phenomena highlighted in the theoretical section also arise when using spectral algorithms. This shows that these behaviors are not mere mathematical artifacts stemming from the increased complexity of PABM relative to DCBM, but that they do occur in practice and are observable in real-world settings.
>    -   Moreover, we conducted additional experiments, where we vary the embedding dimension $d$ and report the results in the table below (we intend to include a figure with these results in the revised version). Under the same parameter settings as in Figure 1, we observed that the performance of *pabm* and *osc*  improves significantly as the dimension increases from $d=3$ to $d=6$, after which it reaches a plateau. In contrast, the performance of the *sbm* and *dcbm* variants remains unchanged with increasing $d$.
>
> | Embedding dimension $d$ | 3    | 5    | 7    | 9    | 11   |
> |-------------------------|------|------|------|------|------|
> | **sbm**                 | 0.63 | 0.65 | 0.65 | 0.64 | 0.65 |
> | **dcbm**                | 0.69 | 0.69 | 0.67 | 0.68 | 0.66 |
> | **pabm**                | 0.39 | 0.86 | 0.89 | 0.94 | 0.97 |
> | **osc**                 | 0.66 | 0.98 | 0.99 | 0.99 | 0.97 |
>
> **Table**: Accuracy of different spectral clustering variants as a function of the embedding dimension $d$.
> Experiments are done with the same model parameters as Figure 1a of the paper ($n=900$ vertices in $k=3$ communities, and we used $\xi = 1$ and $c = 1$). Standard errors are typically less than 0.01 and thus not shown.
>
>
> ### Question
> 1.  Consider a PABM with homogeneous interactions and $k$ clusters of equal-size $n/k$. Because the quantity $\lambda_{i}^{\rm in}$ and $\lambda_i^{\rm out}$ are normalized, the expected number of edges within a cluster is $\frac{n}{k} (\frac{n}{k}-1)p \approx \left( \frac{n}{k} \right)^2 p$ while the number of edges between any two clusters is $\left( \frac{n}{k} \right)^2 q$. Therefore, when $p=q$, the expected edge density is the same within and across clusters. This is what we mean by the absence of an edge-density signal: there is no signal at the \textit{global level} as the difference in expected edge densities between clusters equals zero. However, even when $p=q$, PABM retains a cluster-identifying signal at the \textit{local level}, encoded in the individual edge probabilities $P_{ij}$. This local structure enables cluster recovery in PABM, in contrast to SBM and DCBM, which lack such a local signal under equal edge densities.
> To distinguish the cluster assignments of two vertices $i$ and $j$, global and local signals play different roles. When $p \ne q$, the global signal can be leveraged across all models by simply comparing the number of neighbors that $i$ and $j$ have in each of the $k$ communities. In contrast, to use the much weaker local signal (only present in PABM), one needs to compare the full interaction vectors $P\_{i \\cdot} = (P_{iu})\_{u \\in [n] }$ and $P\_{j \\cdot}  = (P_{ju})\_{u \\in [n] }$, which encode the individual edge probabilities between $i, j$ and all other vertices.

---

> > ### Comment · Reviewer_V4fn · 2025-08-05
> >
> > Thanks for your detailed response. The new experiment addresses my concern on that front. Regarding point 1, I still don’t think the authors have fully acknowledged that these findings are, to some extent, anticipated. When $p = q$, it is natural for the signal to shift to the popularity parameters $\lambda_i$; more broadly, there is an inherent trade-off between community structure and degree heterogeneity. I understand that the authors have worked out the exact nature of this trade-off; the non-monotonic behavior they demonstrate theoretically and corroborate empirically is perhaps unexpected.
> >
> > Overall, I believe these results will interest a subset of the community, and I will raise my score.

---

### Official Review · Reviewer_fUf2 · 2025-07-02

**Clarity:** 2
**Significance:** 2
**Originality:** 2
**Rating:** 4
**Confidence:** 3

**Summary:**

This work studies the problem of graph clustering—the purpose is to let intra-cluster edge-density to be higher than inter-cluster edge-density. Unlike models such as SBM and DCBM, this work focuses on a more general model PABM. PABM allows vertices to be more popular inside their clusters and unpopular with vertices in other clusters.

The primary contribution of this work is to characterize the clustering complexity under PABM via the optimal error rate. The analysis shows that clustering difficulty not only depends on the gap between $p$ and $q$ (the expected intra-cluster and inter-cluster connection probabilities), but also exhibits nuanced variation across vertices—manifesting a more intricate form compared to DCBM.

In addition, this work experimentally examines on the effectiveness of different spectral clustering methods. Traditional spectral algorithms often decompose the adjacency matrix $A$ into a low-dimensional embedding matrix $P$ and a random noise matrix $X$: $A=P+X$, and then use $k$ eigenvectors of $P$ to do clustering. However, to better leverage richer structure present in PABM, recent works incorporate $k^2$ eigenvectors of $P$ instead. The experiments in this work provide evidence that these recent methods surpass the classical approaches in this context.

**Questions:**

* Section 2 is overly dense with mathematical notation and formulas; adding more intuition or high-level discussion (e.g., for Theorem 1 and 2) would significantly improve readability and accessibility.

* Could you clarify your technical contributions more precisely? The literature on optimal clustering error rates in the SBM and related models is extensive. It would be helpful to clearly state which existing techniques your work builds upon and which aspects represent novel contributions.

**Ethical Concerns:**

["NO or VERY MINOR ethics concerns only"]

**Final Justification:**

I think the paper is borderline.

**Limitations:**

Yes

**Quality:**

3

**Strengths And Weaknesses:**

Strengths:

* The paper provides a theoretical analysis of the optimal clustering error rate of a more general random model PABM. The analysis appears to be non-trivial.

Weaknesses:

* The paper emphasizes that their result implies "cluster recovery remains possible in PABM even when traditional edge-density signals (the gap between $p$ and $q$) vanish, provided intra- and inter-cluster popularity coefficients differ." However, I find this somewhat misleading. The PABM model introduces many additional parameters, and even when the gap between $p$ and $q$ disappears, node popularity parameters (such as $\lambda_i^{\text{in}}$, $\lambda_i^{\text{out}}$, etc.) still influence the expected number of intra- and inter-cluster edges. As a result, these parameters can implicitly encode edge-density signals, even in the absence of a direct $p$-$q$ gap.

2. There is no theoretical analysis of spectral clustering methods under the PABM model—only heuristic arguments are provided. In particular, the experimental section on spectral methods appears disconnected from the theoretical results on the optimal error rate in PABM.

3. The main technical contribution of the paper is unclear to me.

---

> ### Author Rebuttal · Authors · 2025-07-29
>
> Dear Reviewer fUf2,
>
> We thank you for your time in evaluating our submission and we are grateful for your comments. Please find below responses to the weaknesses and questions raised in your review.
>
> Weaknesses:
>
> 1.  - Consider a PABM with homogeneous interactions and $k$ clusters of equal-size $n/k$. Because the quantities $\lambda_{i}^{\rm in}$ and $\lambda_i^{\rm out}$ are normalized, the expected number of edges within a cluster is $\frac{n}{k} (\frac{n}{k}-1)p \approx \left( \frac{n}{k} \right)^2 p$ whereas the number of edges between any two clusters is $\left( \frac{n}{k} \right)^2 q$. Therefore, when $p=q$, the expected edge density is the same within and across clusters. This is what we mean by the absence of an edge-density signal: there is no signal at the **global level** as the difference in expected edge densities between clusters equals zero. However, even when $p=q$, PABM retains a cluster-identifying signal at the **local level**, encoded in the individual edge probabilities $P_{ij}$. This local structure enables cluster recovery in PABM, in contrast to SBM and DCBM, which lack such a local signal under equal edge densities.
>     - To distinguish the cluster assignments of two vertices $i$ and $j$, global and local signals play different roles. When $p \ne q$, the global signal can be leveraged across all models by simply comparing the number of neighbors that $i$ and $j$ have in each of the $k$ communities. In contrast, to use the much weaker local signal (only present in PABM), one needs to compare the full interaction vectors $P\_{i \\cdot} = (P_{iu})\_{u \\in [n] }$ and $P\_{j \\cdot}  = (P_{ju})\_{u \\in [n] }$, which encode the individual edge probabilities between $i, j$ and all other vertices.
>
> 2. In the numerical section, we illustrate two surprising phenomena discussed in the theoretical section: the non-monotonic behavior of the error with respect to edge density, and the ability to recover clusters even when $p=q$. While it would have been possible to use a greedy algorithm to approximate the MLE, we opted for spectral methods because of their widespread use and established effectiveness for clustering in block models. In particular, our experiments demonstrate that the phenomena highlighted in the theoretical section also arise when using spectral algorithms. This shows that these behaviors are not merely mathematical artifacts stemming from the increased complexity of PABM relative to DCBM, but do occur in practice and are observable in real-world settings.
>
> 3. Please refer to our response to Question 2 for a detailed explanation of the technical difficulties involved.
>
> Question.
>
> 1. We acknowledge that Section 2 is dense with mathematical notation. However, this level of technical detail is unavoidable given the greater complexity of PABM compared to SBM and DCBM. To help for readability, we present our main results in the introduction for the homogeneous versions of the model, and we include several illustrative examples in Section 2.4. We hope that these choices make the material more accessible to readers.
>
> 2. We acknowledge that we did not include an overview of the proof techniques (partly due to space constraints) and we will add it in the revised version. While the overall structure of the proofs for Theorems 1 and 2, which establish the optimal error rate, is similar to that used for SBM and DCBM, the PABM setting introduces additional technical complexity that requires a more refined analysis.
>    - For the lower-bound, a first challenge is to address the minimum over all permutations in the definition of the error loss. Hence, rather than directly examining $\inf_{\hat{z}} \mathbb{E}[ n^{-1} \mathrm{loss}(z^{\*}, \hat{z}) ]$, we follow previous work such as [Zhang \& Zhou 2016] and [Gao, Ma, Zhang, Zhou 2018] and focus on a sub-problem $\inf_{ \hat{z} \in \mathcal{Z} } \mathbb{E}[ n^{-1} \mathrm{loss}(z^{\*}, \hat{z}) ]$, where $\mathcal{Z} \subset [k]^n$ is chosen such that $\mathrm{loss}(z^{\*}, \hat{z}) = \mathrm{Ham}( z^{\*}, \hat{z} )$ for all  $z^{\*}, \hat{z} \in \mathcal{Z}$. The idea is that this sub-problem is simple enough to analyze, while still capturing the hardness of the original clustering problem. Next, we use a result from [Dreveton, Gözeten, Grossglauser, Thiran 2024] to show that the Bayes risk $\inf_{\hat{z}_i} \mathbb{P} ( \hat{z}_i \ne z_i^{\*})$ for the misclustering of a single vertex $i$ is asymptotically $e^{-(1+o(1)) \mathrm{ Chernoff}(i,z^{\*})}$. (We refer to answer of Question 1 of reviewer 2qS9 for an overview of this result.) We acknowledge that this part follows to some extent from existing and established techniques in the literature on optimal error rates in block models and mixture models.
>    - The proof of the upper bound is however more involved, and required a new approach. It begins, similarly to prior work on block models, by upper-bounding $\mathbb{E} [ \mathrm{loss}(z^{\*}, \hat{z}) ]$ (where $\hat{z}$ is the MLE) by $\sum_m \mathbb{P}( \mathrm{Ham}(z^{\*}, \hat{z}) = m)$. Thus, the core difficulty relies in upper-bound the quantities $\mathbb{P}(L(z) > L(z^{\*}))$ for any $z$ such that $\mathrm{Ham}(z^{\*}, z) = m$ (where $L(z)$ denotes the likelihood of $z$ given an observation of $A$). This is more challenging in PABM than in SBM and DCBM. Indeed, unlike in SBM (and to some extent DCBM), the likelihood ratio $\frac{L(z)}{L(z^{\*})}$ cannot be easily simplified. As for SBM and DCBM, we rely on Chernoff bounds to obtain $\mathbb{P}(L(z) > L(z^{\*})) \le \mathbb{E} [ e^{t \log (L(z) /  L(z^{\*}) } ]$ for any $t>0$. But, in SBM and DCBM, one can use $t = 1/2$ and obtain clean exponential bounds whose terms are $\exp(- \theta_u \theta_v (\sqrt{p}-\sqrt{q})^2)$. For PABM, the optimal $t$ to use depends intricately on the misclassified set $\{ u \colon z_u \ne z_u^*\}$, and thus on $z$ itself. To address this, we adopt a more refined approach: we decompose the upper bound into three components $T_1(t)$, $T_2(t)$, and $T_3(t)$, and select a tailored value of $t$ for each labeling $z$. This additional complexity distinguishes our analysis from earlier work and reflects the greater structural richness of PABM compared to SBM and DCBM.

---

> > ### Comment · Reviewer_fUf2 · 2025-08-07
> >
> > Thank you for your response. I have a clearer understanding of the intuition behind theorem 1 and 2 now.  I will raise my score.

---

### Comment · Area_Chair_W6s8 · 2025-08-01
**The time to start author-reviewer discussions**

Dear all reviewers,

The author rebuttal period has now concluded, and authors' responses are
available for the papers you are reviewing. The Author-Reviewer Discussion
Period has started, and runs until August 6th AoE.

Your active participation during this phase is crucial for a fair and
comprehensive evaluation. Please take the time to:

- Carefully read the author responses and all other reviews.
- Engage in a constructive dialogue with the authors, clarifying points,
  addressing misunderstandings, and discussing any points of disagreement.
- Prioritize responses to questions specifically addressed to you by the authors.
- Post your initial responses as early as possible within this window to
  allow for meaningful back-and-forth discussion.

Your insights during this discussion phase are invaluable.
Thank you for your continued commitment to the NeurIPS review process.

Bests,
Your AC

---

### Decision · Program_Chairs · 2025-09-17

**Decision:**

Accept (poster)

**Comment:**

This paper presents a strong theoretical contribution to the field of graph
clustering by establishing the information-theoretic limits of community
detection under the Popularity-Adjusted Block Model (PABM). The key insight
is that PABM, a generalization of the well-studied SBM and DCBM, allows for
cluster recovery even when traditional global edge-density signals are
absent, by leveraging more nuanced local differences in node connectivity
patterns. The authors provide tight, matching upper and lower bounds on the
optimal clustering error rate, a technically sound and significant result.

Initially, the reviewers, while recognizing the technical solidity, raised
several important concerns. These included the clarity of the novel
technical contributions compared to prior work, the potentially misleading
framing of "clustering without edge density signals," and a perceived
disconnect between the theoretical analysis based on the maximum likelihood
estimator and the experiments using spectral methods.

However, the authors provided rebuttals that thoroughly addressed these
points. They clarified that their main claim refers to the absence of a
global density signal, effectively shifting the focus to richer, local
structural information. Crucially, they detailed the novel technical
aspects of their proofs, particularly for the upper bound, which requires a
more refined analysis than for simpler models. Their arguments successfully
demonstrated that the empirical results, while using a practical algorithm,
serve to validate the non-intuitive theoretical phenomena uncovered by
their analysis. This comprehensive response led multiple reviewers to raise
their scores, resulting in a strong consensus for acceptance.

And the current submission is filled with mathematics, while missing
important explanations that tell readers the motivations, importance and
understandings of these theoretical analyses.

Overall, it should be noted that the current state of this submission has
not yet met the quality demand of NeurIPS. The final version must soundly
integrate these explanations and additional clarifications to meet the
quality of NeurIPS publication.